# Factors Influencing Generalization in Chaotic Dynamical Systems

## Abstract

Many real-world systems exhibit chaotic behaviour, for example: weather, fluid dynamics, stock markets, natural ecosystems, and disease transmission. While chaotic systems are often thought to be completely unpredictable, in fact there are patterns within and across that experts frequently describe and contrast qualitatively. We hypothesise that given the right supervision / task definition, representation learning systems will be able to pick up on these patterns, and successfully generalize both in- and out-of-distribution (OOD). Thus, this work explores and identifies key factors which lead to good generalization. We observe a variety of interesting phenomena, including: learned representations transfer much better when fine-tuned vs. frozen; forecasting appears to be the best pre-training task; OOD robustness falls off very quickly outside the training distribution; recurrent architectures generally outperform others on OOD generalization. Our findings are of interest to any domain of prediction where chaotic dynamics play a role.

## 1 Introduction

There are many reasons to be interested in understanding and predicting behaviour of chaotic systems. For example, the current climate crisis is arguably the most important issue of our time. From atmospheric circulation and weather prediction to economic and social patterns, there are chaotic dynamics in many data relevant to mitigate impact and adapt to climate changes. Most natural ecosystems exhibit chaos; a better understanding of the mechanisms of our impact on our environment is essential to ensuring a sustainable future on our planet. The spread of information in social networks, many aspects of market economies, and the spread of diseases, all have chaotic dynamics too, and of course these are not isolated systems - they all interact in complex ways, and the interaction dynamics can also exhibit chaos.

This makes chaotic systems a compelling challenge for machine learning, particularly representation learning: Can models learn representations that capture high-level patterns and are useful across other tasks? Which losses, architectures, and other design choices lead to better representations? These are some of the questions which we aim to answer. Our main contributions are:

- The development of a lightweight evaluation framework, **ValiDyna**, to evaluate representations learned by deep-learning models in new tasks, new scenarios, and on new data.

- The design of experiments using this framework, showcasing its usefulness and flexibility.

- A comparative analysis of 4 popular deep-learning architectures using these experiments.

**Table 1:** Summary of the generalisation results. S, C and F stand for the tasks of Supervised featurisation, Classification, and Forecasting. $A \nrightarrow B$ and $A \rightarrow B$ indicate strict (see section 5.2) and loose (see section 5.3) feature-transfer from task $A$ to task $B$. All runs generalise in-distribution. ✓ and − indicate whether or not the model-run pair achieves OOD generalization in the final task.

| model | S | C | F | $S \nrightarrow C$ | $F \nrightarrow C$ | $S \nrightarrow F$ | $C \nrightarrow F$ | $F \rightarrow S$ | $F \rightarrow C$ | $C \rightarrow S$ | $C \rightarrow F$ |
|---|---|---|---|---|---|---|---|---|---|---|---|
| GRU | ✓ | ✓ | − | ✓ | ✓ | − | − | ✓ | ✓ | ✓ | − |
| LSTM | ✓ | ✓ | − | ✓ | ✓ | − | − | ✓ | ✓ | ✓ | − |
| Transformer | ✓ | ✓ | − | ✓ | ✓ | − | − | ✓ | − | − | − |
| N-BEATS | − | − | − | − | − | − | − | − | − | − | − |

## 2 RELATED WORK

Many works have studied factors influencing generalization for deep networks; see Maharaj (2022) for review, and Arjovsky (2021) for OOD specifically. To our knowledge, ours is the first such analysis for data exhibiting chaotic dynamics.

Our work relies on that of Gilpin (2021), which presents a dataset of dynamical systems that show chaotic behaviour under certain conditions. They benchmark statistical and deep-learning models typically used with time series for a variety of tasks including forecasting and dataset-transfer, and highlight some connections between model performance and chaotic properties.

Although not directly addressing chaos, the intersection of physics-informed and dynamical systems literature with representation learning holds relevance for chaotic dynamics, e.g. Raissi et al. (2019) show how to train models whose predictions respect the laws of physics, by employing partial differential equations as regularisation. Yin et al. (2022) propose a framework to learn contextual dynamics by decomposing the learned dynamical function into two components that capture context-invariant and context-specific patterns.

As AI systems are increasingly deployed in the real world, researchers have increasingly noted shortcomings of standard practice (i.e. performance on validation/test set) for comprehensively evaluating learned representations. An increasing number of evaluation frameworks have been proposed to help address this, e.g. Gulrajani & Lopez-Paz (2021) propose model selection algorithms and develop a framework (DomainBed) for testing domain/OOD generalisation. Of particular relevance, Wang et al. (2020) discuss the difference between generalisation to new data domains and to new ODE parameters in the context of dynamical systems. They show that ML techniques generalise badly when the parameters of a test system/data are not included in the train set (extrapolation).

## 3 DATA

Our data is generated using `dysts`, a Python library of 130+ chaotic dynamical systems published by Gilpin (2021). In `dysts`, each dynamical system can be integrated into a trajectory with any desired initial condition, length and granularity, thus allowing to generate an unlimited number of trajectories. It can also generate trajectories of similar time scales across different chaotic systems. See Figure 1 for examples and Figure A9 for further examples.

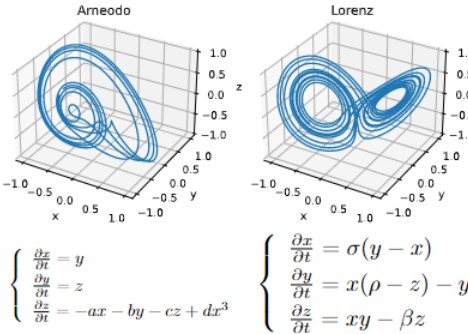

**Figure 1:** Sample trajectories from two related chaotic attractors. Both systems have two 'lobes'; Arneodo (**left**) has a characteristic shell shape with one lobe inside the other, while Lorenz (**right**) shows a characteristic butterfly shape with lobes at an angle to one another. This is the kind of high-level pattern experts describe for many real-world chaotic systems, which we hypothesize representation learning systems could pick up on.

### 3.1 THE DATA GENERATION PROCESS

We sample data from each dynamical system by picking different initial conditions. This leads to trajectories that are sufficiently different from each other, but representative of the underlying chaotic system. However, `dysts` relies on numerical ODE solvers to generate trajectories, which could fail due to numerical instabilities when the initial condition is too extreme. To avoid that, we generate the default trajectory for each system, compute the component-wise minima and maxima, and use a percentage $p$ of the resulting intervals to sample random initial conditions for that system. In addition to the properties of the trajectory, the parameters of this process are the random seed and the percentage $p$ of the observed initial condition range to be used for sampling.

## 3.2 OUR DATASETS AND THEIR PARAMETERS

We generate three sets of data for all our experiments: training, validation, and test. The training set is used to optimise the model weights. The validation set is used for early stopping and learning rate adaptation (see Appendix A.3), and to measure in-distribution generalisation. The test set is used to measure OOD generalisation.

The train and validation sets come from the same data distribution, while the test set comes from a larger distribution containing the former. All the sets contain trajectories with the same length (5 periods) and granularity (50 points per period). The parameters used to generate the data can be found in Table A2. We choose to only include dysts systems of 3 dimensions in the datasets (i.e. 100 out of 131 systems, cf. Table A3) to avoid adapting models for variable input dimensions, and for faster training. The default trajectory from each included system can be seen in Figure A9.

## 4 THE VALIDYNA EVALUATION FRAMEWORK

The exploratory and comparative nature of this work results in the need to have a common experimental framework for consistency and configurability across experiments, including different tasks and combinations of losses. We present **ValiDyna**, an open-source, lightweight framework built on top of Pytorch and Lightning . It is built with extensibility in mind, so that new model architectures, metrics and training objectives can be easily added. The framework saves a large amount of code repetition and complex indexing/references, e.g. in multi-task experiments.

### 4.1 TASKS

ValiDyna currently includes three tasks on learned representations from with time series data:

1. **(Task S) Self-supervised featurisation** (aka **feature extraction**) involves extracting features from time series such that similar time series have similar features, similarity defined as coming from the same dynamical system. We use a triplet margin loss, which takes 3 *features* as input: that of an anchor time series $a$, a *positive* series $p$ similar to it, and a *negative* (dissimilar) series $n$:

$$L_{\text{triplet}}(a, p, n) = max(d(a, p) - d(a, n) + m, 0)$$

where $d$ is a distance metric (euclidean in our case) and $m$ is the margin of tolerance, i.e. the minimum difference between the positive and negative distances for the loss to be non-zero. The number of features to be extracted and the margin value are the main parameters of this task.

2. **(Task C) Classification** involves predicting a single discrete class for each time series, in our case the chaotic system from which it came. We use cross-entropy loss to measure how close the model's output is to the true class. The main parameter is the data-dependent number of classes.

3. **(Task F) Forecasting**, perhaps the most popular task for time-series, involves predicting the future values of a time series based on its past values. We use the mean squared error (MSE) loss. Although the number of time steps in the past and in the future need not necessarily be fixed, we do so due to N-BEATS' architecture (cf. Section 4.2). Thus, this task is parameterised by the number $T_{\text{in}}$ of time steps that are input to the model, and the number $T_{\text{out}}$ of time steps output by the model.

Each of these tasks is implemented in ValiDyna as a separate Lightning module (`[Slice]Featuriser`, `Classifier` and `Forecaster`) that wraps around a model architecture to allow for easy training and metric logging. All such modules log the corresponding loss during training for all data sets, while the `Classifier` module additionally logs the classification accuracy.

### 4.2 MODEL ARCHITECTURES

ValiDyna currently includes 4 machine learning architectures often used for temporal data:

- **GRU** (Cho et al., 2014) and **LSTM** (Hochreiter & Schmidhuber, 1997): these Recurrent Neural Networks (**RNNs**) are likely the most popular ML architectures to be used for time series as they allow crunching a series of variable size into a fixed-size representations.
- **Transformer** (Vaswani et al., 2017): an attention-based architecture that achieves state-of-the-art performance for seq2seq, and has replaced LSTMs in many time series tasks.

- **N-BEATS** (Oreshkin et al., 2020): a purely deep neural state-of-the-art forecasting architecture based on residual blocks. Originally written in TensorFlow, we provide a PyTorch implementation based on that of Herzen et al. (2021).

The main challenge of our multi-task setup involves adapting the model architectures above for tasks they were not originally built for. The most straightforward way is to use the architecture (or part of it) as a feature extractor, and then attach a classification or forecasting head. For **RNNs**, we consider the outputs of the last layer as the "features". For the **Transformer**, we use its encoder as a feature extractor, and completely discard the decoder. For **N-BEATS**, we choose the concatenation of the forecast neural basis expansions of all blocks as the "features".

To ensure fair comparisons, the framework makes it easy to ensure a fixed number of features ($N_{features}$) across model architectures. To accomplish this, we insert a simple linear layer with $N_{features}$ output units between the vanilla feature extractor and the task-specific heads. These models are implemented in ValiDyna as sub-classes of a `MultiTaskModel` with all of the functionality above, as shown in Figure 2. For further details on the framework, see Appendix A.1

## 5 EXPERIMENTS

We now present some experiments that help us better understand how different deep-learning models represent chaotic data, and showcase the ValiDyna framework. For all experiments, we show a table with the metrics achieved by the various model runs during the last epoch of training, and show the corresponding training curves in the Appendix. The configuration values used across experiments are listed in Appendix A.3 for the sake of transparency and reproducibility.

### 5.1 RANDOM SAMPLING

In this baseline experiment, we measure the dependence of model performance on the specific set of trajectories used for training. We construct 5 subsets (using the random seeds 0 to 4) sampling 75 % of the available trajectories in each set without replacement. For each sub-sampled set, we train each model architecture for each of the 3 tasks. The results in Table 2 also serve as a performance baseline for the more complex experiments that follow.

Table 2 shows the following: N-BEATS consistently performs poorly for featurisation and classification (an alternate featuriser/forecaster decomposition could potentially perform better); OOD forecasting generalisation is bad overall; GRU, LSTM and Transformer perform better on the validation set than on the train set (possibly an effect of dropout regularisation).

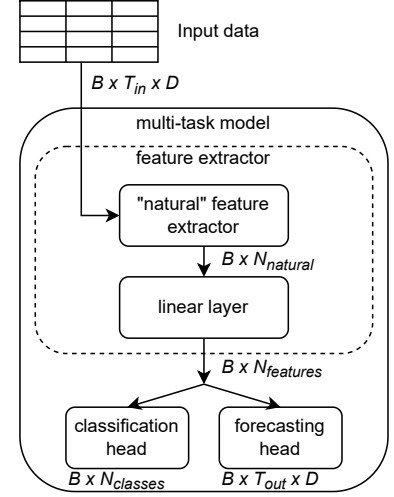

**Figure 2:** Schematics of a multi-task model. $B$=batch size, $D$=space dimension, $T$=number of time steps. $N_{natural}$ depends on model hyper-parameters, but $N_{features}$ can be chosen. N-BEATS needs no separate forecasting head.

The training curves in Figure A2 show that: different seeds result in different training times although final performance is stable; test curves are noise-like (no improvement) for N-BEATS on all tasks, and for forecasting with all models.

### 5.2 STRICT FEATURE-TRANSFER ACROSS TASKS (FROZEN WEIGHTS)

With this experiment, we seek to evaluate the usefulness of transferring learned representations from one of the three tasks to another. We expect forecasting to rely on implicitly learning the system of a time series to generate better predictions, and hope that pre-training for tasks that do this explicitly will be beneficial. Moreover, since classification and featurisation try to achieve very similar goals, we expect that training a model for one is beneficial for the other.

In each run, we: **1.** pre-train a model for one task (the pre-training task); **2.** freeze the parameters associated to the feature extractor (cf. Figure 2); **3.** train for another task (the main task). Note that featurisation cannot be a main task as freezing the featuriser results in no learning.

**Table 2:** Random sampling experiment: final metric means and standard deviations aggregated over the 5 different random sampling seeds. See Table A4 for the full results per sampling seed. We highlight the best mean value obtained on each set.

N-BEATS performs poorly for classification and featurisation. GRU is consistently among the best performers, LSTM is close behind. Validation metrics are equal or better than the train ones for all models but N-BEATS. Generalisation to the OOD test set is generally good, except for forecasting.

| *set* | train | validation | test |
|---|---|---|---|
| GRU | **.12**±.009 | **.09**±.008 | **.10**±.008 |
| LSTM | **.12**±.009 | .10±.009 | .11±.009 |
| Transformer | .25±.005 | .21±.001 | .21±.002 |
| N-BEATS | .48±.035 | .48±.034 | 2.60±1.800 |

**(a) (Task S)** Supervised featurisation loss (↓)

| *set* | train | validation | test |
|---|---|---|---|
| GRU | **5.1**±.32 | 4.1±.23 | 2.1e+8±2.9e+7 |
| LSTM | 5.2±.5 | **4**±.34 | 2.4e+8±2.3e+3 |
| Transformer | 9.1±.94 | 8.6±.98 | 1.9e+8±9.3e+6 |
| N-BEATS | 7±0.34 | 7.2±.34 | **1.5e+8**±1.6e+8 |

**(b) (Task F)** Forecasting loss ×1000 (↓)

| *set* | train | validation | test |
|---|---|---|---|
| GRU | **.28**±.036 | **.21**±.032 | **.33**±.033 |
| LSTM | .31±.052 | .20±.042 | .36±.046 |
| Transformer | .52±.050 | .50±.055 | .65±.052 |
| N-BEATS | 2.30±.039 | 2.30±.041 | .70±.052 |

**(c) (Task C)** Classification loss (↓)

| *set* | train | validation | test |
|---|---|---|---|
| GRU | **92**±1.1 | **95**±0.9 | **93**±.89 |
| LSTM | 91±1.5 | 94±1.2 | 92±1.2 |
| Transformer | 85±1.6 | 86±2.1 | 85±2.1 |
| N-BEATS | 40±0.6 | 40±.66 | 39±.75 |

**(d) (Task C)** Classification accuracy % (↑)

**Table 3:** Feature-freeze experiment: final task metrics as a function of the pre-training task. We highlight the best metric value obtained for each model-dataset pair.

In general, pre-training on other tasks results in a worse performance. Interestingly, the representations learned by Transformer during forecasting seem to transfer well to classification. Freezing the feature extractor entirely seems to prevent learning.

| *set* | train | | | validation | | | test | | |
|---|---|---|---|---|---|---|---|---|---|
| *pre-task* | C | S | – | C | S | – | C | S | – |
| GRU | 37 | 35.5 | **4.8** | 28 | 27.7 | **3.8** | **1.35e+8** | 1.42e+8 | 1.6e+8 |
| LSTM | 41.4 | 40.8 | **4.75** | 32.1 | 32 | **3.62** | **1.32e+8** | 1.42e+8 | 1.56e+8 |
| Transformer | 13 | 21.4 | **8.51** | 12.6 | 19.8 | **7.96** | 1.16e+8 | **9.48e+7** | 1.18e+8 |
| N-BEATS | 170 | 236 | **6.22** | 169 | 237 | **6.36** | 1.12e+8 | 8.3e+7 | **3.88e+7** |

**(a) (Task F)** Forecasting loss ×1000 (↓)

| *set* | train | | | validation | | | test | | |
|---|---|---|---|---|---|---|---|---|---|
| *pre-task* | S | F | – | S | F | – | S | F | – |
| GRU | 1.64 | 1.15 | **.178** | 1.31 | .913 | **.117** | 1.41 | 1.08 | **.267** |
| LSTM | 1.96 | 1.45 | **.354** | 1.61 | 1.13 | **.245** | 1.76 | 1.26 | **.378** |
| Transformer | 1.44 | **.448** | .529 | 1.11 | .471 | **.448** | 1.19 | .655 | **.648** |
| N-BEATS | 2.28 | 3.37 | **2.03** | 2.28 | 3.38 | **2.04** | 21.6 | **3.45** | 43.7 |

**(b) (Task C)** Classification loss (↓)

| train | | | validation | | | test | | |
|---|---|---|---|---|---|---|---|---|
| S | F | – | S | F | – | S | F | – |
| 52.6 | 65.7 | **94.7** | 63.2 | 73.5 | **97** | 61.9 | 72 | **95.5** |
| 44.4 | 57.4 | **89.5** | 54.6 | 67.8 | **93.5** | 53.1 | 66.3 | **91.5** |
| 55.8 | **87.9** | 84.4 | 66.1 | 87.5 | **87.8** | 64.7 | 85.6 | **85.7** |
| 38.7 | 19.1 | **46.1** | 38.6 | 19 | **46** | 37.3 | 18.5 | **44.7** |

**(c) (Task C)** Classification accuracy % (↑)

Table 3 shows that in almost all cases, pre-training on other tasks results in a worse performance. The only exception is when pre-training Transformer for forecasting and then classifying, but even then GRU and LSTM still achieve better classification accuracy with no pre-training. Figure A3 shows that pre-training for other tasks puts models in a better initial position during the main training, but then performance stops improving and ends up worse than without pre-training. Given this initial performance boost, we speculate that learned features are useful across tasks, but freezing the featuriser is too extreme and prevents learning during the main training phase.

## 5.3 PROBING FOR OTHER TASKS (FINE-TUNING)

The goal of this experiment is to better understand how training a model for one task impacts its performance on other tasks. The flexibility of our framework allows to write this experiment in a simple manner, by implementing a new "prober" metric that actively treats a task module A as if it were the module for task B, and logs metrics for B.

It is to be noted that to probe for some tasks, a model must have already been pre-trained for that task. For instance, one cannot probe for classification or forecasting during featurisation training. However, we can probe for featurisation while training for other tasks, as the featuriser is used for those tasks (except N-BEATS which does not train its featuriser layer). A side-effect of this experiment is that it solves the main limitation of the previous one, by allowing the transfer of features across tasks without freezing any component of the model.

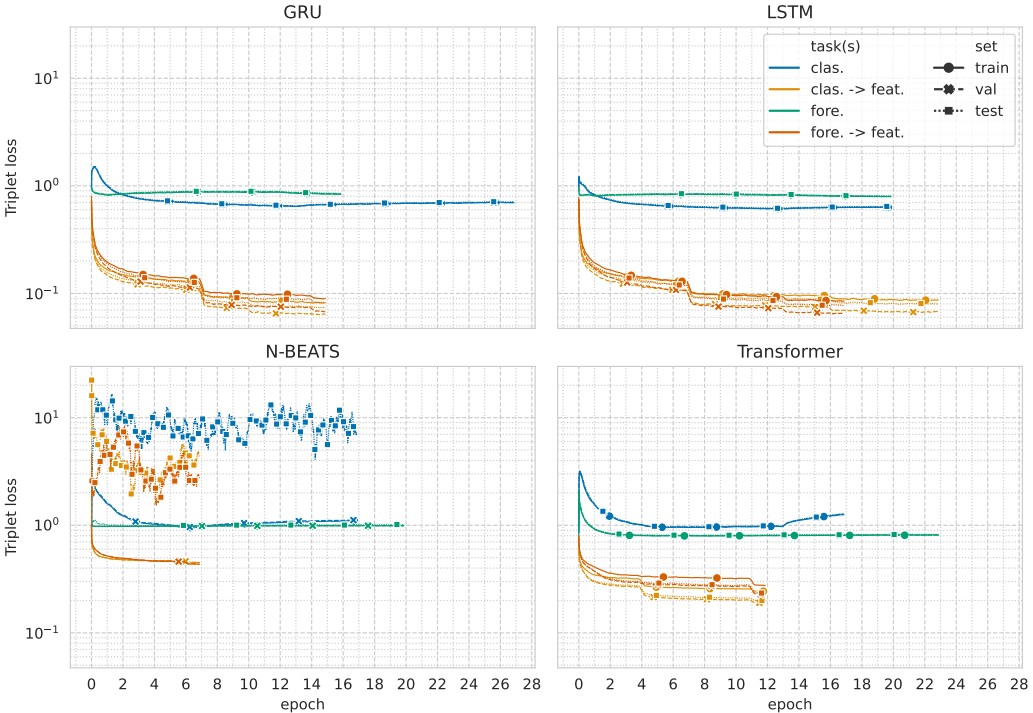

**Figure 3:** Prober experiment: training curves per model and pre/main task combination. A running average of length 500 (roughly a quarter of an epoch) is used for readability.
The best task performance is obtained when the model is being trained for that specific task. Metrics of tasks different to the training task seem to stay stable during training.

First, we look at task performance using pre-training, extending the observations of Section 5.2. Table 4 show that: the best classification performance is with forecasting pre-training; all models except N-BEATS perform forecasting at least as well on the train/val sets with classification pre-training; featurisation loss is generally better with pre-training than without (cf. Table 2a baseline).

Then, for task probing, Table 4 shows that the classification metrics are consistently bad during the training of other tasks: accuracy $\approx 1\%$ (the random baseline), and the loss is an order of magnitude higher than its counterparts on all sets. This is not surprising. Consider $C \to F$, as the featuriser is updated during forecasting training, it is no longer compatible with the classification head.

Considering the evolution of metrics during training, Figure 3 shows that the performance of task A when training for task B is initially good, but either collapses after one or two epochs, or stays stable. This agrees with our theory that the featuriser and task-specific heads become incompatible.

In summary, this experiment shows that pre-training on other tasks can be greatly beneficial for another task, likely because it places the model parameters in a region of space that is easier to optimise. However, the update of the featuriser weights during training renders the previously-trained task-specific heads useless as they cannot adapt to the new features that they receive.

**Table 4:** Prober experiment: final metrics per pre-training task, training task, model and dataset. The best metric value is highlighted for each model-set pair.
Forecasting features transfer well to classification and the reverse is often true. Classification and forecasting features transfer decently to featurisation (cf. baseline in Table 2a).

| set | train | | | | validation | | | | test | | | |
|---|---|---|---|---|---|---|---|---|---|---|---|---|
| pre-task | C | F | – | | C | F | – | | C | F | – | |
| task | S | S | C | F | S | S | C | F | S | S | C | F |
| GRU | **.08** | .09 | .7 | .84 | **.06** | .07 | .7 | .84 | **.07** | .08 | .71 | .85 |
| LSTM | .09 | **.08** | .63 | .79 | .07 | **.06** | .63 | .8 | .08 | **.08** | .64 | .81 |
| Transformer | **.24** | .28 | 1.2 | .81 | **.19** | .23 | 1.3 | .82 | **.20** | .24 | 1.3 | .82 |
| N-BEATS | .45 | **.44** | 1.1 | .99 | .45 | **.44** | 1.1 | .99 | 4.1 | 2.6 | 8.5 | **1** |

**(a) (Task S)** Supervised featurisation loss ($\downarrow$)

| set | train | | | | validation | | | | test ($\div 10^6$) | | | |
|---|---|---|---|---|---|---|---|---|---|---|---|---|
| pre-task | – | C | F | | – | C | F | | – | C | F | |
| task | F | F | C | S | F | F | C | S | F | F | C | S |
| GRU | **.46** | .46 | 28 | 28 | .36 | **.34** | 28 | 28 | 16 | 16 | **13** | 16 |
| LSTM | **.42** | .42 | 27 | 27 | .33 | **.32** | 27 | 28 | 16 | 16 | **13** | 17 |
| Transformer | .81 | **.80** | 28 | 28 | .76 | **.75** | 28 | 28 | 13 | 12 | 13 | **9.9** |
| N-BEATS | **.60** | 2 | 28 | 42 | **.61** | 2 | 28 | 42 | **2.7** | 8.8 | 13 | 13 |

**(b) (Task F)** Forecasting loss $\times 100$ ($\downarrow$)

| set | train | | | | validation | | | | test | | | |
|---|---|---|---|---|---|---|---|---|---|---|---|---|
| pre-task | – | C | F | | – | C | F | | – | C | F | |
| task | C | S | F | C | C | S | F | C | C | S | F | C |
| GRU | .21 | 4.7 | 4.8 | **.12** | .15 | 4.7 | 4.8 | **.08** | .29 | 4.7 | 4.8 | **.23** |
| LSTM | .29 | 4.7 | 4.6 | **.15** | .21 | 4.7 | 4.6 | **.09** | .35 | 4.7 | 4.6 | **.21** |
| Transformer | .45 | 6.3 | 4.7 | **.39** | .44 | 6.3 | 4.7 | **.37** | .63 | 6.3 | 4.7 | **.58** |
| N-BEATS | 2 | 4.8 | 4.7 | **1.9** | 2.1 | 4.8 | 4.7 | **1.9** | 38 | 4.8 | **4.7** | 74 |

**(c) (Task C)** Classification loss ($\downarrow$)

| set | train | | | | validation | | | | test | | | |
|---|---|---|---|---|---|---|---|---|---|---|---|---|
| pre-task | – | C | F | | – | C | F | | – | C | F | |
| task | C | S | F | C | C | S | F | C | C | S | F | C |
| GRU | 93.8 | 1 | 1 | **96.3** | 96.3 | 1 | 1.01 | **97.9** | 94.6 | 1.01 | 1 | **96.6** |
| LSTM | 91.4 | 1 | 1 | **95.6** | 94.5 | 1.01 | 1.01 | **97.8** | 92.8 | 1 | 1 | **96.5** |
| Transformer | 85.6 | 0.97 | 1 | **89.2** | 87.9 | 0.97 | 1 | **90.3** | 85.7 | 0.97 | 0.99 | **88.3** |
| N-BEATS | 45.9 | 1.84 | 1 | **50.2** | 45.7 | 1.84 | 1 | **50.1** | 44.3 | 1.79 | 0.99 | **48.5** |

**(d) (Task C)** Classification accuracy % ($\uparrow$)

## 5.4 Few-Shot learning

With this experiment, we hope to better understand how our models adapt to a distribution shift consisting of a new environment with a new chaotic system in it. We focus on the dynamical system **SprottE**. We consider two sets of systems: a set of 4 toy chaotic systems with simple equations similar to SprottE's (Sprott, 1994): SprottA, SprottB, SprottC and SprottD; and a set of 4 systems with more complex differential equations: Arneodo, Lorenz, Sakarya and QiChen. We show the default trajectory and differential equations of each system in Figure A5.

The experiment is set as follows. In some cases, pre-train a model on one set of 4 systems (similar or different), then add SprottE to the data and train fully. In other cases, train models directly on one set of 5 systems (SprottE + similar/different). Runs are identified by the similarity of the other systems (similar vs. different), and by whether SprottE is included ("no" during pre-training, "no→yes" after pre-training, and "yes" when SprottE is there from the beginning).

Since we care about model performance on SprottE in particular, we introduce new metrics: for forecasting, MSE loss only on SprottE i.e. S-MSE; for classification, sensitivity i.e. true positive rate (TPR) and specificity i.e. true negative rate (TNR) of SprottE vs the other classes; for featurisation, the standard deviation of the features extracted from SprottE series. Note that these metrics can only be tracked when SprottE is included in the data (i.e. not during pre-training).

Table 5 confirms our choice of similar and different systems: classification accuracy and featurisation loss are better for different systems (i.e. easier to differentiate), while forecasting loss is better for similar (i.e. reusable representations). Pre-training does not have a significant impact on classification or forecasting metrics, but is better for featurisation, in particular for different systems. This could be due to the triplet margin loss needing more samples to be optimised than the two others.

Figure A6 mainly shows that convergence is faster for pre-trained models, as expected. There does not seem to be any relationship between SprottE feature standard deviation and performance.

In this experiment, we also visualise the features learned by the models using 2D PCA projections. Figure 4 shows features learned under the feature extraction task for the 4 different architectures. Full plots for all settings in Figure A11.

**Table 5:** Few-shot learning experiment: final metrics per training task, system similarity ($=$ or $\neq$), and SprottE status, averaged over the train/validation/test sets. See Table A5 for the full table. Forecasting MSE is significantly better for the similar systems. Featurisation loss and classification accuracy are better for the different systems.

| *SprottE?* | no | | no→yes | | yes | |
|---|---|---|---|---|---|---|
| *systems* | $\neq$ | $=$ | $\neq$ | $=$ | $\neq$ | $=$ |
| GRU | .27 | .74 | .26 | .76 | .44 | .79 |
| LSTM | .38 | 2 | .29 | .57 | .81 | 1.6 |
| Transformer | .63 | 3 | .42 | 2.3 | .54 | 2.4 |
| N-BEATS | 2.7 | 3.8 | 2 | 4.5 | 5.2 | 4.1 |

**(a) (Task S)** Feat. loss $\times 100$ ($\downarrow$)

| *SprottE?* | no | | no→yes | | | | yes | | | |
|---|---|---|---|---|---|---|---|---|---|---|
| *systems* | $\neq$ | $=$ | $\neq$ | | $=$ | | $\neq$ | | $=$ | |
| | MSE | MSE | MSE | S-MSE | MSE | S-MSE | MSE | S-MSE | MSE | S-MSE |
| GRU | 3.3 | 1.1 | 2.3 | 1.6 | .78 | 1.3 | 1.8 | 1.1 | .87 | 1.1 |
| LSTM | 4.4 | 1.2 | 2.2 | 1.4 | 1.1 | 1.7 | 2.7 | 1.6 | 1.1 | 1.3 |
| Transformer | 2 | .54 | 1.7 | .94 | .7 | 1.2 | 2.4 | 1.4 | .98 | 1.3 |
| N-BEATS | 1.8 | .68 | 2.1 | 1.4 | .63 | .97 | 2.2 | 1.4 | .91 | 1.2 |

**(b) (Task F)** Forecasting metrics $\times 1000$ ($\downarrow$) (S-MSE = SprottE MSE)

| *SprottE?* | no→yes | | yes | |
|---|---|---|---|---|
| *systems* | $\neq$ | $=$ | $\neq$ | $=$ |
| GRU | 1.10 | .080 | 1.5 | 0.47 |
| LSTM | 0.61 | 0.68 | 1.4 | 0.91 |
| Transformer | 0.88 | 0.45 | 0.8 | 0.81 |
| N-BEATS | 1.50 | 1.60 | 1.8 | 0.44 |

**(c) (Task S)** Featurisation std

| *SprottE?* | no | | no→yes | | | | | yes | | | | |
|---|---|---|---|---|---|---|---|---|---|---|---|---|
| *systems* | $\neq$ | $=$ | $\neq$ | | | $=$ | | $\neq$ | | | $=$ | |
| | acc | acc | TNR | TPR | acc | TNR | TPR | acc | TNR | TPR | acc | TNR | TPR | acc |
| GRU | 100 | 100 | 100 | 100 | 100 | 100 | 100 | 99 | 100 | 100 | 100 | 100 | 100 | 100 |
| LSTM | 100 | 97 | 100 | 100 | 100 | 100 | 100 | 98 | 100 | 100 | 100 | 100 | 100 | 100 |
| Transformer | 100 | 98 | 100 | 100 | 100 | 100 | 100 | 99 | 100 | 100 | 100 | 100 | 100 | 100 |
| N-BEATS | 99 | 97 | 99 | 93 | 97 | 100 | 99 | 98 | 100 | 97 | 95 | 100 | 99 | 94 |

**(d) (Task C)** Classification metrics % ($\uparrow$)

**Figure 4:** Comparison of the features extracted by the 4 different architectures. Note the effective low dimensionality of the N-BEATS features compared to the others. In these examples and in general, recurrent architectures LSTM and GRU appear to have the most separable learned features.

A very noticeable result is that the two principal axes of the PCA projections of the features generated by N-BEATS always explain at least 95% of the feature variance (we call this value 'r'). We speculate that, although the feature extractor of N-BEATS has an output dimension of 32, its effective number of degrees of freedom (i.e. its effective capacity) is much lower, around 2, which would explain why features for all systems are mixed in a linear or "V" shape of most PCA projections.

### 5.5 COMBINING TASK LOSSES

In this experiment, we explore optimising the three task losses simultaneously. Concretely, we implement a new `SliceModule` whose loss is a weighted mean of all task losses. In particular, we consider forecasting as a main task, and use the other losses to explicitly enforce the learning of a series' system, by setting the weights $L_{\text{total}} = \alpha L_{\text{MSE}} + (1-\alpha)\left(L_{\text{triplet}} + L_{\text{cross}}\right)$.

Figure A10 shows no evident benefit from enforcing shared representations across tasks.

### 6 LIMITATIONS AND FUTURE DIRECTIONS

**Limitations:** Despite ValiDyna's configurability and extensibility, it has some limitations inherent in choosing an experimental scope:

- It is built with time series data at its core, and would not work with other types of data such as static images (e.g. forecasting is time series-specific).
- While adding new architectures is simple, adapting new models to multi-task learning requires some expertise.

- There are many ways to extract features for some of the model architectures (e.g. for N-BEATS), and the current version only support a single scheme per model architecture.
- It only supports training models on a single dataset, and has no way of distinguishing between different training environments corresponding for example to different ecosystems.

There are also limitations in our experiments:

- They were done for a particular timescale of data, and might differ substantially when considering coarser or finer granularities. This is a challenge in general for climate data as some events occur in large timescales and others in smaller ones, and climate models must account for both.
- Out-of-distribution generalisation was only explored in the context of extrapolation initial condition of trajectories.
- Although the dysts library allows generating noisy trajectories, the data used in the experiments is free of noise, while real-world measurements often contain noise.

**Future directions:** Apart from evaluating further architectures and adding new losses to the existing framework, there are a few ways in which we would like to expand our framework,. While initial experiments with meta-learning were not promising (this is why we did not focus the framework around them), adding the ability to do multi-dataset losses such as meta-learning losses is something we would like to pursue. We hypothesize that the benefits of this approach might require massively multi-environment settings to show. We would also like to exploring generative modelling and the role it can play in encouraging good representations.

We would also like to explore further experimental settings, such as mixing different data timescales, dynamical system parameters (e.g. changing $\beta$ in Lorenz), and trajectory noise levels.

Finally, an area of ongoing work is to perform a comparison with real ecological measurements and see how the results differ from the synthetic case.

## 7 CONCLUSION

In summary, we present an experimental analysis of factors influencing generalization for data exhibiting chaotic dynamics. To do so, we built a configurable and extensible model evaluation framework called Validyna. Using Validyna, we constructed and ran five experiments — random sampling, transfer learning with frozen weights, fine-tuning with unfrozen weights (probing), few-shot learning, multi-task loss — to better understand the quality of representations learned by four popular machine learning architectures — GRU, LSTM, Transformer, N-BEATS — on three tasks — feature extraction, classification, forecasting — for in- and out-of-distribution generalization.

**Takeaways**. Summarizing our extensive experiments, the main takeaways from our work are:

- All four model architectures generalise poorly to an unseen data distribution for the forecasting task. This is likely due to the chaotic nature of our data.
- Our feature extractor for N-BEATS performs very poorly, while the others perform better.
- All four model architectures are robust to data sub-sampling in the sense that their final performance is stable, but training times can vary considerably.
- Dropout seems to be an effective regularizer for in- and out-of-distribution generalisation.
- Learned representations can transfer well across tasks, especially from forecasting to classification, but not when the feature extraction module is frozen.
- There is no straightforward relationship between optimising for the triplet or cross-entropy loss, although they try to achieve a very similar goal.
- The cross-entropy loss and the classification accuracy of a model do not necessarily follow each other when models are optimised for other losses.
- There is no evident benefit from enforcing shared representations across tasks.

These results provide insights and starting points for future research in representation learning of chaotic dynamical systems.

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

# A   APPENDIX

## A.1   ADDITIONAL DETAILS ON VALIDYNA EVALUATION FRAMEWORK

### A.1.1   DATA PROCESSING

Entire trajectories cannot be fed to the models due to N-BEATS' limitation of requiring a fixed number of input and output time steps $T_{\text{in}}$ and $T_{\text{out}}$. Also, task-specific data i.e. positive and negative examples for the (S) featurisation task need to be constructed. And, although the generation process is transparent to the different scales of system values (cf. Section 3.1), the generated trajectories still need to be scaled.

To address these issues, we: **1.** compute the component-wise minima and maxima of the train trajectories of each system; **2.** use them to scale the trajectories per system in all 3 sets; **3.** map each (scaled) trajectory of length $N$ into the list of all the possible contiguous slices of length $T_{\text{in}} + T_{\text{out}}$; **4.** split each such slice in two parts of length $T_{\text{in}}$ and $T_{\text{out}}$; **5.** attach the system name which generated it, encoded as a number. Thus, a single data sample in our setting is a triplet $(\boldsymbol{X}_{\text{in}}, \boldsymbol{X}_{\text{out}}, c)$ i.e. model input for all 3 tasks, the true future for forecasting, and the class label for classification.

In addition, given a batch of anchor time series, the `Featuriser` module can retrieve a batch of one positive and one negative example per anchor.

### A.1.2   FRAMEWORK ARCHITECTURE, EXTENSIBILITY

Introducing a new multi-task model to the framework is straightforward, as it suffices to implement a new `MultiTaskModel` class and adapt the model to all available tasks. Adding new metrics to be tracked for specific tasks or experiments is also very straightforward, and the process is briefly explained in the experiments of Sections 5.3 and 5.4. Creating variations of current tasks can also be done quickly, e.g. adding a Forecaster that optimises the mean average error (MAE) instead of the MSE is trivial. However, introducing a new task can be complicated, as it requires: adapting all existing multi-task models to it; implementing the core training objective in a new `SliceModule`; adapting `MultiTaskDataset` and the data processing pipeline if new kinds of data are necessary. See Figure A1 for a high-level class diagram.

## A.2   EXAMPLE OF EXPERIMENT CODE

With our framework, one run for the strict feature-transfer experiment could be simply written as:

```
# Python 3.9 pseudocode
model: MultiTaskModel = GRU(...)
dataset: MultiTaskDataset = ...
# Pre-train for classification
classifier = Classifier(model)
classifier.fit(dataset)
# Freeze feature extractor weights
model.freeze_featurizer()
# Train model for forecasting
forecaster = Forecaster(model)
forecaster.fit(dataset)
```

## A.3   EXPERIMENTAL SETTINGS

We use the values $T_{\text{in}} = T_{\text{out}} = 5$ for data processing and for the models. For each dataset, we shuffle the data and use batches of size 1024. After every training batch, we compute the metrics of interest for a random validation and test batch. All models are optimised using PyTorch's AdamW optimiser, with a starting learning rate of 0.01 that is divided by 5 when the validation loss does not improve, with a patience of 1 epoch. All training procedures include early stopping, so that training stops when the validation loss stops decreasing, with a patience of 3 epochs. A maximum of 100

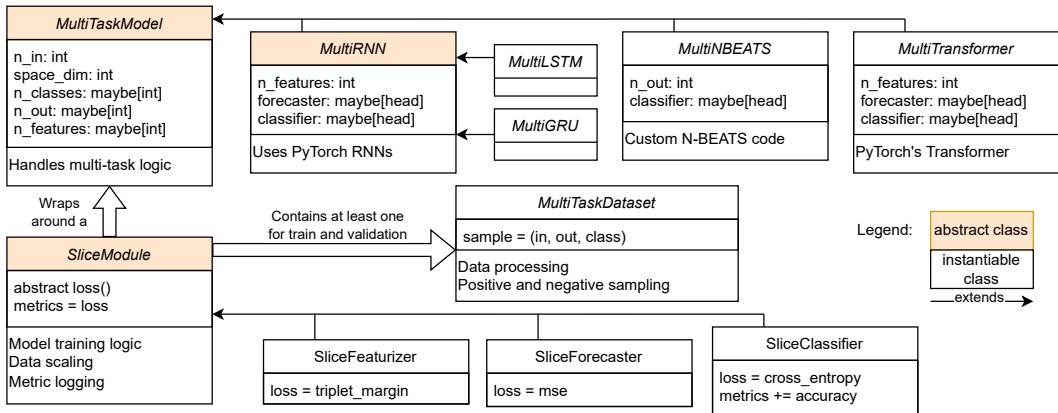

**Figure A1:** Class diagram of the ValiDyna Framework.

training epochs is allowed, but no training run attains it. Each model training is run deterministically, by setting Lightning's random seed to 2022.

We also want to make sure that our models have a comparable representative power, and we use the number of parameters as a proxy. Table A1 shows the number of parameters obtained for each model using the hyper-parameters that follow. All models use the value $N_{\text{features}} = 32$. All forecasting and classification heads in our models are simple feed-forward neural networks with 3 hidden layers of width equal to $N_{\text{features}} = 32$, and ReLU activations after each hidden layer. For N-BEATS, we use 4 stacks of 4 blocks each, a neural-basis expansion dimension of 4, and the fully-connected network of each block has 4 hidden layers of 8 units each and ReLU activation. For the two RNNs, we use a dropout probability of 0.1 and 2 layers, with GRU having 30 hidden units per layer and LSTM 26. The Transformer uses 4 encoder layers, each having a feed-forward dimension of 6, as well 4 attention heads, an embedding dimension of 16, and a dropout probability of 0.1 in the feed-forward and self-attention networks. The margin of the triplet loss is equal to the default of 1.

**Table A1:** The number of parameters of each model architecture as used in the experiments. We consider the number of parameters used in the featuriser and in total (including any task-specific heads). The total parameter count is very stable (around 11-12k), while there is slightly more variability for the featuriser one (around 7-9k).

| *model* | GRU | LSTM | Transformer | N-BEATS |
|---|---|---|---|---|
| # params (featuriser) | 8730 | 8840 | 6816 | 7856 |
| # params (total) | 11933 | 11915 | 12079 | 11136 |

## B  EXTRA TABLES

**Table A2:** Parameters used to generate trajectories for each data set. Theoretically, the trajectories from the train and validation sets come from the same distribution, while those from the test set come from a larger distribution containing the other. *see Section 3.1.

| *parameter* | random seed | # trajectories | $p^*$ | # periods | # $\frac{\text{points}}{\text{period}}$ | # points |
|---|---|---|---|---|---|---|
| train | 0 | 80 | 5 % | 5 | 50 | 250 |
| validation | 1 | 20 | 5 % | 5 | 50 | 250 |
| test | 2 | 30 | 10 % | 5 | 50 | 250 |

**Table A3:** The distribution of attractor dimensions in dysts. Most attractors have dimension 3.

| *dimension* | 3 | 4 | 5 | 6 | 10 |
|---|---|---|---|---|---|
| *count* | 100 | 19 | 2 | 3 | 7 |

**Table A4:** Random sampling experiment: full final metrics per random sampling seed.

| *set* | train | | | | | validation | | | | | test | | | | |
|---|---|---|---|---|---|---|---|---|---|---|---|---|---|---|---|
| *seed* | 0 | 1 | 2 | 3 | 4 | 0 | 1 | 2 | 3 | 4 | 0 | 1 | 2 | 3 | 4 |
| GRU | .103 | .113 | .116 | .116 | .128 | .0812 | .0917 | .092 | .0916 | .103 | .0897 | .0991 | .102 | .1 | .113 |
| LSTM | .128 | .107 | .124 | .119 | .131 | .105 | .0838 | .101 | .0949 | .107 | .114 | .0936 | .109 | .106 | .115 |
| Transformer | .243 | .257 | .255 | .255 | .255 | .207 | .203 | .206 | .206 | .206 | .216 | .209 | .213 | .213 | .213 |
| N-BEATS | .455 | .529 | .444 | .495 | .46 | .458 | .531 | .447 | .497 | .464 | 2.92 | 2.19 | 5.37 | .526 | 1.8 |

**(a) (Task S)** Supervised featurisation loss (↓)

| *set* | train | | | | | validation | | | | | test | | | | |
|---|---|---|---|---|---|---|---|---|---|---|---|---|---|---|---|
| *seed* | 0 | 1 | 2 | 3 | 4 | 0 | 1 | 2 | 3 | 4 | 0 | 1 | 2 | 3 | 4 |
| GRU | 5.12 | 4.94 | 5.71 | 4.96 | 4.98 | 4.06 | 3.9 | 4.49 | 4.01 | 4.02 | 1.96e+8 | 1.72e+8 | 2.36e+8 | 2.36e+8 | 2.36e+8 |
| LSTM | 5.34 | 4.64 | 5.07 | 5.94 | 4.89 | 4.2 | 3.65 | 3.98 | 4.54 | 3.87 | 2.36e+8 | 2.36e+8 | 2.36e+8 | 2.36e+8 | 2.36e+8 |
| Transformer | 9.14 | 10.7 | 8.86 | 8.36 | 8.36 | 8.65 | 10.3 | 8.39 | 7.86 | 7.86 | 1.89e+8 | 2.03e+8 | 1.82e+8 | 1.82e+8 | 1.82e+8 |
| N-BEATS | 6.79 | 6.97 | 6.85 | 7.65 | 6.96 | 7.07 | 7.13 | 7.02 | 7.83 | 7.14 | 4.12e+8 | 5.49e+7 | 3.46e+7 | 1.6e+8 | 7.84e+7 |

**(b) (Task F)** Forecasting loss (↓)

| *set* | train | | | | | validation | | | | | test | | | | |
|---|---|---|---|---|---|---|---|---|---|---|---|---|---|---|---|
| *seed* | 0 | 1 | 2 | 3 | 4 | 0 | 1 | 2 | 3 | 4 | 0 | 1 | 2 | 3 | 4 |
| GRU | .274 | .274 | .332 | .232 | .265 | .206 | .198 | .259 | .171 | .203 | .32 | .297 | .379 | .3 | .336 |
| LSTM | .307 | .317 | .35 | .366 | .232 | .223 | .241 | .267 | .28 | .172 | .347 | .381 | .394 | .387 | .283 |
| Transformer | .558 | .488 | .49 | .598 | .49 | .486 | .549 | .443 | .557 | .443 | .642 | .7 | .603 | .711 | .603 |
| N-BEATS | 2.34 | 2.23 | 2.29 | 2.29 | 2.29 | 2.36 | 2.25 | 2.3 | 2.3 | 2.3 | 147 | 101 | 33.8 | 33.8 | 33.8 |

**(c) (Task C)** Classification loss (↓)

| *set* | train | | | | | validation | | | | | test | | | | |
|---|---|---|---|---|---|---|---|---|---|---|---|---|---|---|---|
| *seed* | 0 | 1 | 2 | 3 | 4 | 0 | 1 | 2 | 3 | 4 | 0 | 1 | 2 | 3 | 4 |
| GRU | .919 | .918 | .901 | .93 | .921 | .947 | .948 | .932 | .956 | .948 | .932 | .933 | .916 | .94 | .932 |
| LSTM | .908 | .906 | .896 | .891 | .93 | .94 | .938 | .93 | .926 | .956 | .924 | .922 | .914 | .91 | .942 |
| Transformer | .836 | .855 | .859 | .823 | .859 | .866 | .84 | .883 | .842 | .883 | .847 | .823 | .866 | .824 | .866 |
| N-BEATS | .396 | .413 | .401 | .401 | .401 | .394 | .413 | .402 | .402 | .402 | .384 | .403 | .387 | .387 | .387 |

**(d) (Task C)** Classification accuracy (random baseline of 0.01) (↑)

**Table A5:** Few-shot learning experiment: full final metrics.

| | SprottE? | no | | no->yes | | | | | | yes | | | | | |
|---|---|---|---|---|---|---|---|---|---|---|---|---|---|---|---|
| | attractors | ≠ | = | ≠ | | | = | | | ≠ | | | = | | |
| | metric | acc | acc | TNR | TPR | acc | TNR | TPR | acc | TNR | TPR | acc | TNR | TPR | acc |
| train | GRU | .999 | .997 | 1 | 1 | 1 | .999 | .999 | .996 | 1 | 1 | 1 | 1 | 1 | .998 |
| | LSTM | .998 | .973 | 1 | 1 | .999 | .999 | .997 | .98 | 1 | 1 | .998 | 1 | 1 | .997 |
| | Transformer | .993 | .981 | 1 | 1 | .998 | .999 | .999 | .996 | 1 | 1 | .998 | 1 | 1 | .997 |
| | N-BEATS | .996 | .972 | .995 | .932 | .978 | .997 | .994 | .982 | 1 | .975 | .953 | .998 | .993 | .951 |
| validation | GRU | .999 | .997 | 1 | 1 | 1 | .998 | .999 | .995 | 1 | 1 | 1 | .999 | 1 | .998 |
| | LSTM | .998 | .97 | 1 | 1 | 1 | .997 | .998 | .979 | 1 | 1 | .999 | 1 | 1 | .998 |
| | Transformer | .998 | .983 | 1 | 1 | .999 | .999 | 1 | .997 | 1 | 1 | .999 | .999 | 1 | .997 |
| | N-BEATS | .991 | .966 | .993 | .93 | .971 | .995 | .992 | .975 | .999 | .975 | .948 | .997 | .99 | .945 |
| test | GRU | .994 | .995 | .999 | .997 | .998 | 1 | .997 | .993 | .999 | .999 | .998 | 1 | .996 | .994 |
| | LSTM | .996 | .969 | .999 | .997 | .998 | 1 | .995 | .974 | .999 | .997 | .996 | 1 | .997 | .993 |
| | Transformer | .996 | .977 | 1 | .997 | .998 | 1 | .995 | .992 | .997 | .998 | .996 | 1 | .995 | .993 |
| | N-BEATS | .993 | .961 | .982 | .925 | .962 | .995 | .99 | .974 | .998 | .971 | .934 | .999 | .99 | .936 |

**(a) (Task C)** Classification metrics (↑)

| | SprottE? | no | | no->yes | | | | | | yes | | | | | |
|---|---|---|---|---|---|---|---|---|---|---|---|---|---|---|---|
| | attractors | ≠ | = | ≠ | | = | | ≠ | | = | | | | | |
| | metric | MSE | MSE | MSE | S-MSE | MSE | S-MSE | MSE | S-MSE | MSE | S-MSE | | | | |
| train | GRU | .00196 | .00145 | .0014 | .00139 | .000923 | .000974 | .0011 | .000991 | .0011 | .00103 | | | | |
| | LSTM | .00259 | .00148 | .00142 | .00132 | .00129 | .00146 | .00181 | .00162 | .00124 | .00112 | | | | |
| | Transformer | .000606 | .000567 | .00065 | .000553 | .000592 | .000507 | .00106 | .000915 | .000953 | .000791 | | | | |
| | N-BEATS | .000624 | .00062 | .0009 | .000721 | .000482 | .000438 | .000888 | .000694 | .000718 | .000648 | | | | |
| validation | GRU | .0024 | .00119 | .00148 | .00058 | .000545 | .000363 | .0012 | .000372 | .000686 | .000364 | | | | |
| | LSTM | .00356 | .00119 | .00164 | .000548 | .000845 | .000648 | .00194 | .000737 | .000881 | .000518 | | | | |
| | Transformer | .00137 | .000577 | .0015 | .000396 | .000566 | .000382 | .00198 | .000686 | .00087 | .000485 | | | | |
| | N-BEATS | .00194 | .000808 | .00207 | .000813 | .000618 | .00052 | .00207 | .000868 | .000966 | .000768 | | | | |
| test | GRU | .00568 | .000697 | .00404 | .00282 | .000859 | .00245 | .00317 | .00197 | .000822 | .00191 | | | | |
| | LSTM | .00705 | .000853 | .00343 | .00241 | .00116 | .00308 | .00438 | .0025 | .00103 | .00229 | | | | |
| | Transformer | .00392 | .000463 | .00306 | .00186 | .000946 | .00277 | .00419 | .00258 | .00111 | .00249 | | | | |
| | N-BEATS | .00289 | .00061 | .00325 | .00258 | .000781 | .00194 | .00375 | .00268 | .00104 | .00228 | | | | |

**(b) (Task F)** Forecasting metrics (↓)

| | SprottE? | no | | no->yes | | | | yes | | | | | | | |
|---|---|---|---|---|---|---|---|---|---|---|---|---|---|---|---|
| | attractors | ≠ | = | ≠ | | = | | ≠ | | = | | | | | |
| | metric | L | L | L | σ | L | σ | L | σ | L | σ | | | | |
| train | GRU | .00254 | .00547 | .0022 | 1.07 | .00568 | .795 | .00191 | 1.49 | .0062 | .466 | | | | |
| | LSTM | .00271 | .0163 | .00152 | .605 | .00351 | .673 | .00545 | 1.37 | .0123 | .903 | | | | |
| | Transformer | .00954 | .0265 | .00682 | .853 | .0215 | .442 | .00952 | .769 | .0221 | .788 | | | | |
| | N-BEATS | .019 | .0249 | .00823 | 1.55 | .0339 | 1.57 | .0389 | 1.82 | .0356 | .444 | | | | |
| validation | GRU | .00161 | .00398 | .00166 | 1.08 | .00626 | .803 | .0014 | 1.5 | .0056 | .466 | | | | |
| | LSTM | .00244 | .0178 | .00115 | .607 | .0036 | .685 | .00399 | 1.37 | .0153 | .908 | | | | |
| | Transformer | .00292 | .026 | .00133 | .891 | .0199 | .454 | .00211 | .821 | .0231 | .818 | | | | |
| | N-BEATS | .0293 | .053 | .0154 | 1.55 | .0552 | 1.56 | .0597 | 1.82 | .0468 | .443 | | | | |
| test | GRU | .00395 | .0128 | .00405 | 1.08 | .0108 | .803 | .0098 | 1.5 | .012 | .467 | | | | |
| | LSTM | .00632 | .027 | .00596 | .608 | .01 | .685 | .0148 | 1.37 | .0202 | .907 | | | | |
| | Transformer | .00648 | .0363 | .00454 | .891 | .0271 | .456 | .00462 | .821 | .0255 | .819 | | | | |
| | N-BEATS | .0325 | .037 | .0362 | 1.55 | .0463 | 1.56 | .0576 | 1.82 | .0402 | .443 | | | | |

**(c) (Task S)** Supervised featurisation metrics (↓ L)

## C    EXTRA FIGURES

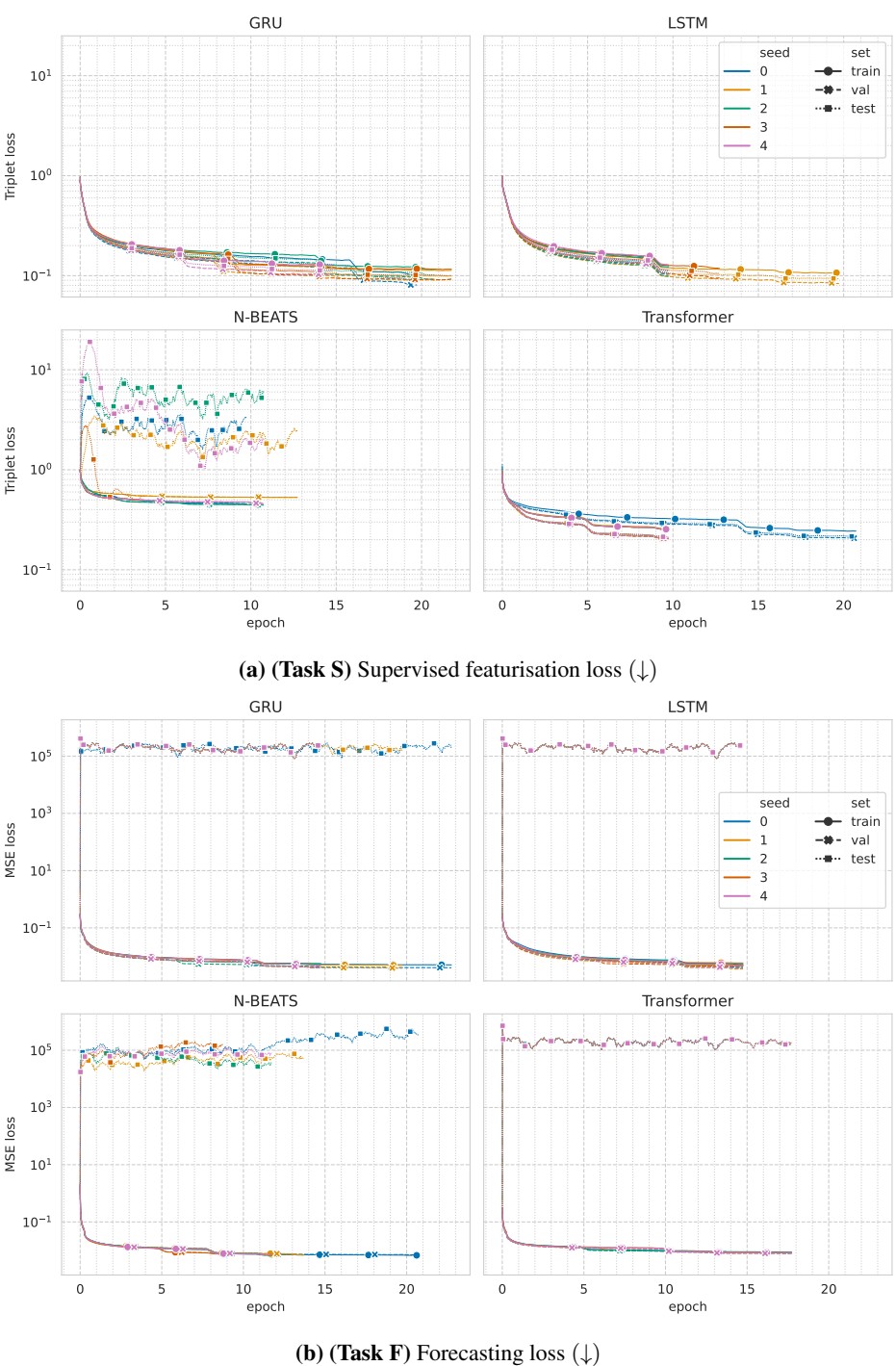

**(a) (Task S)** Supervised featurisation loss (↓)

**(b) (Task F)** Forecasting loss (↓)

**Figure A2:** Random sampling experiment: training curves. A running average of length 700 (roughly half an epoch) is used for readability.

The random seeds don't seem to impact the final performance of the models, but they do impact the training times and speed of convergence.

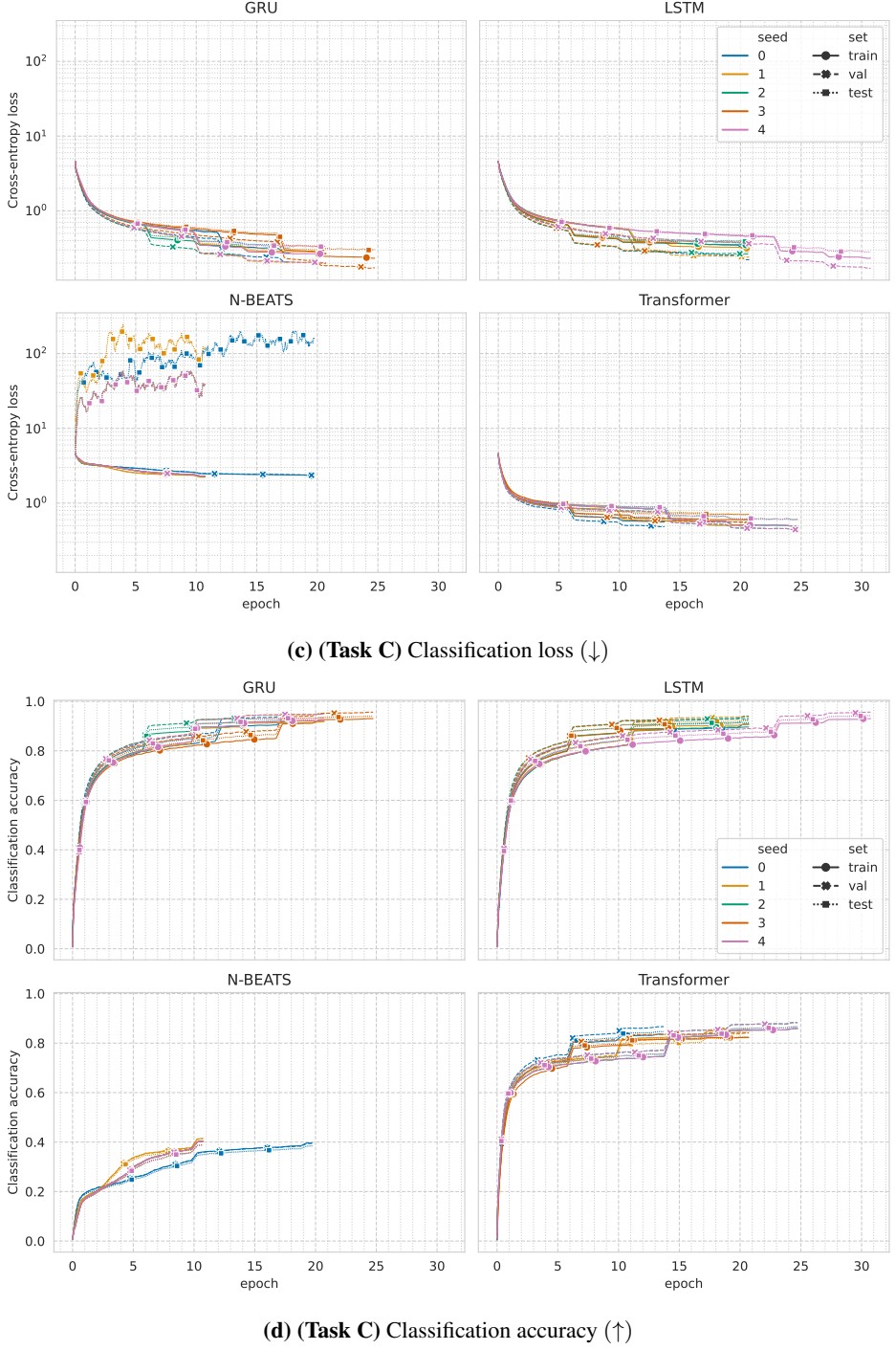

**(c) (Task C)** Classification loss (↓)

**(d) (Task C)** Classification accuracy (↑)

**Figure A2:** Random sampling experiment: training curves (cont.)

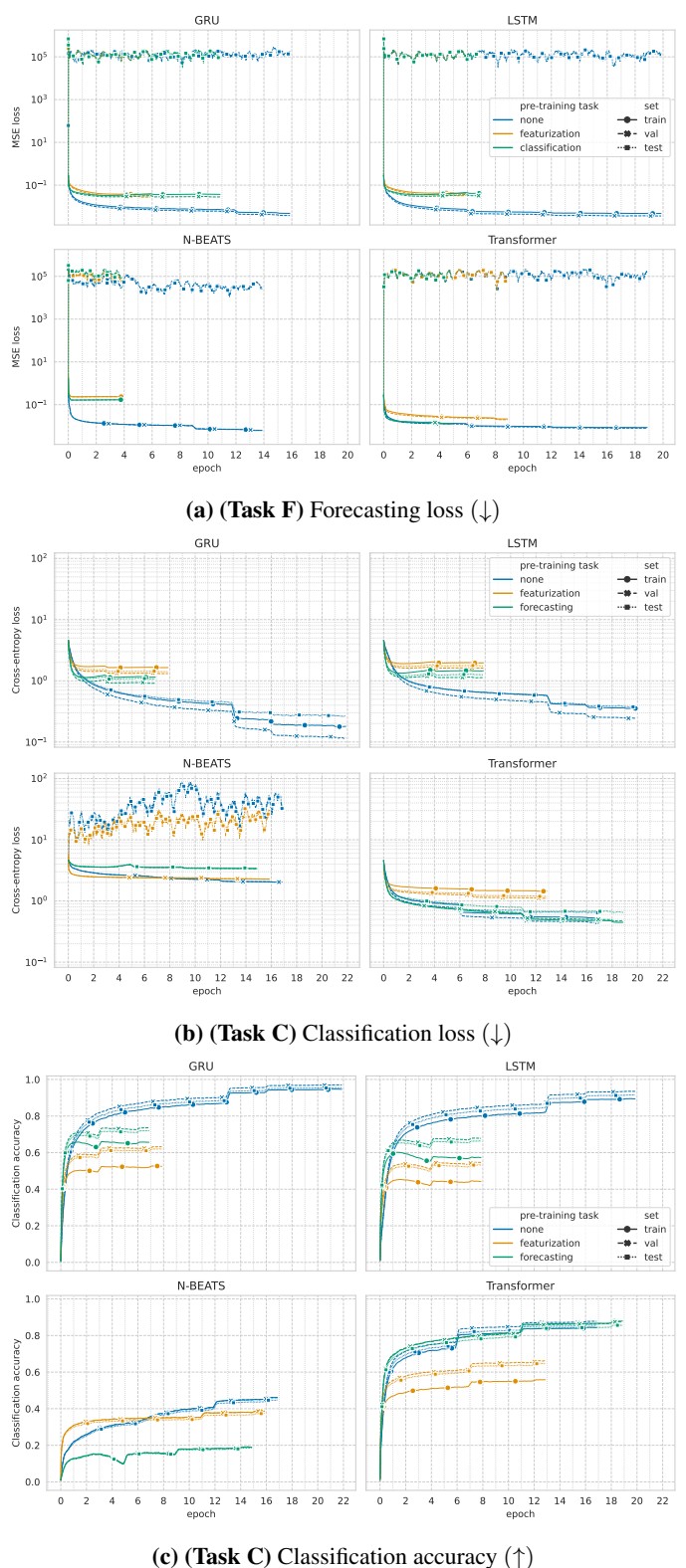

**(a) (Task F)** Forecasting loss (↓)

**(b) (Task C)** Classification loss (↓)

**(c) (Task C)** Classification accuracy (↑)

**Figure A3:** Feature-freeze experiment: training curves for the classification and forecasting tasks. A running average of length 500 (roughly a quarter of an epoch) is used for readability.
Pre-training seems to help at the start of training, but performance quickly plateaus. This is likely due to the frozen featuriser weights preventing learning from happening.

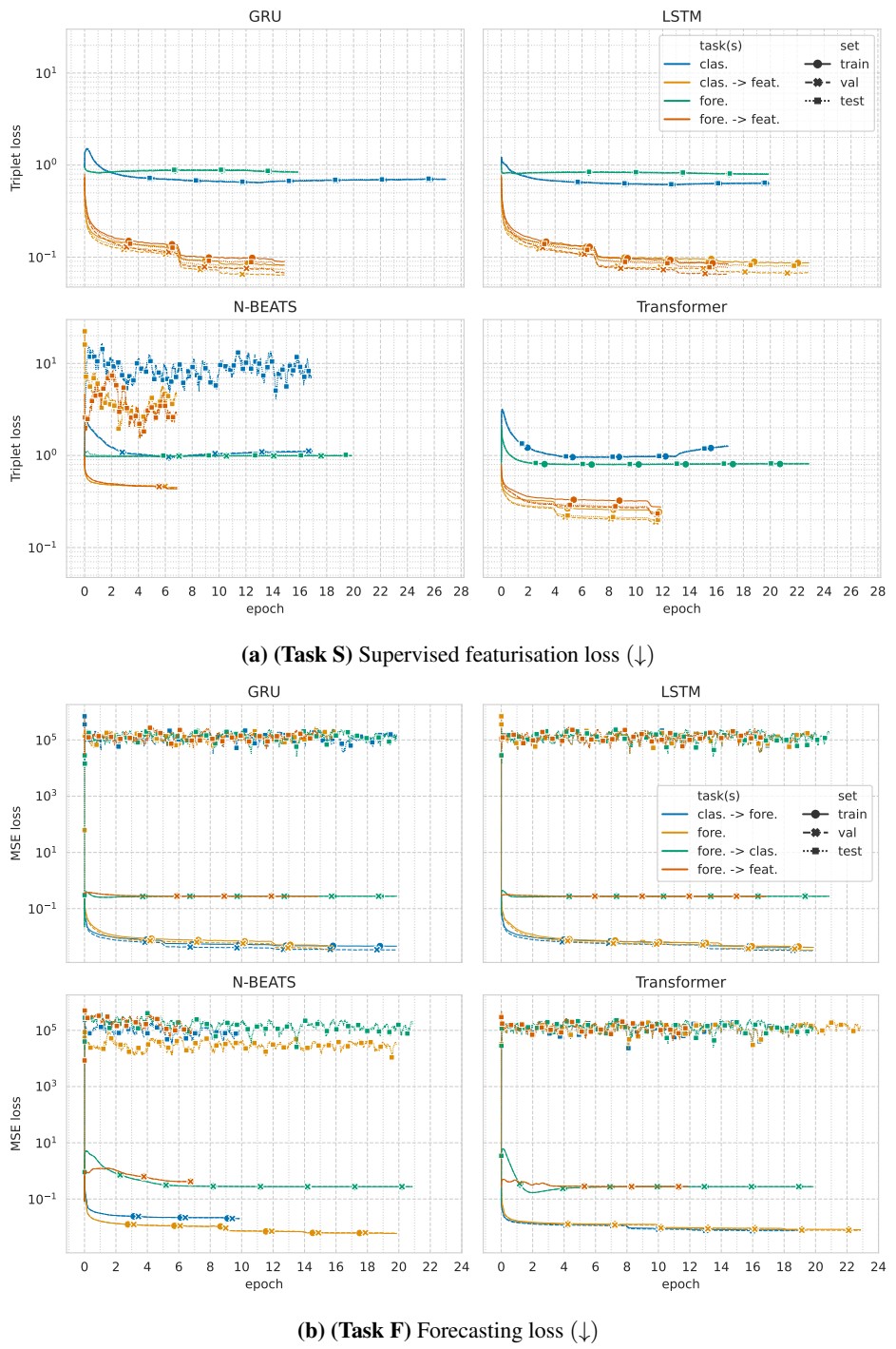

**(a) (Task S)** Supervised featurisation loss ($\downarrow$)

**(b) (Task F)** Forecasting loss ($\downarrow$)

**Figure A4:** Prober experiment: training curves per model and pre/main task combination. A running average of length 500 (roughly a quarter of an epoch) is used for readability.

The best task performance is obtained when the model is being trained for that specific task. Metrics of tasks different to the training task seem to stay stable during training.

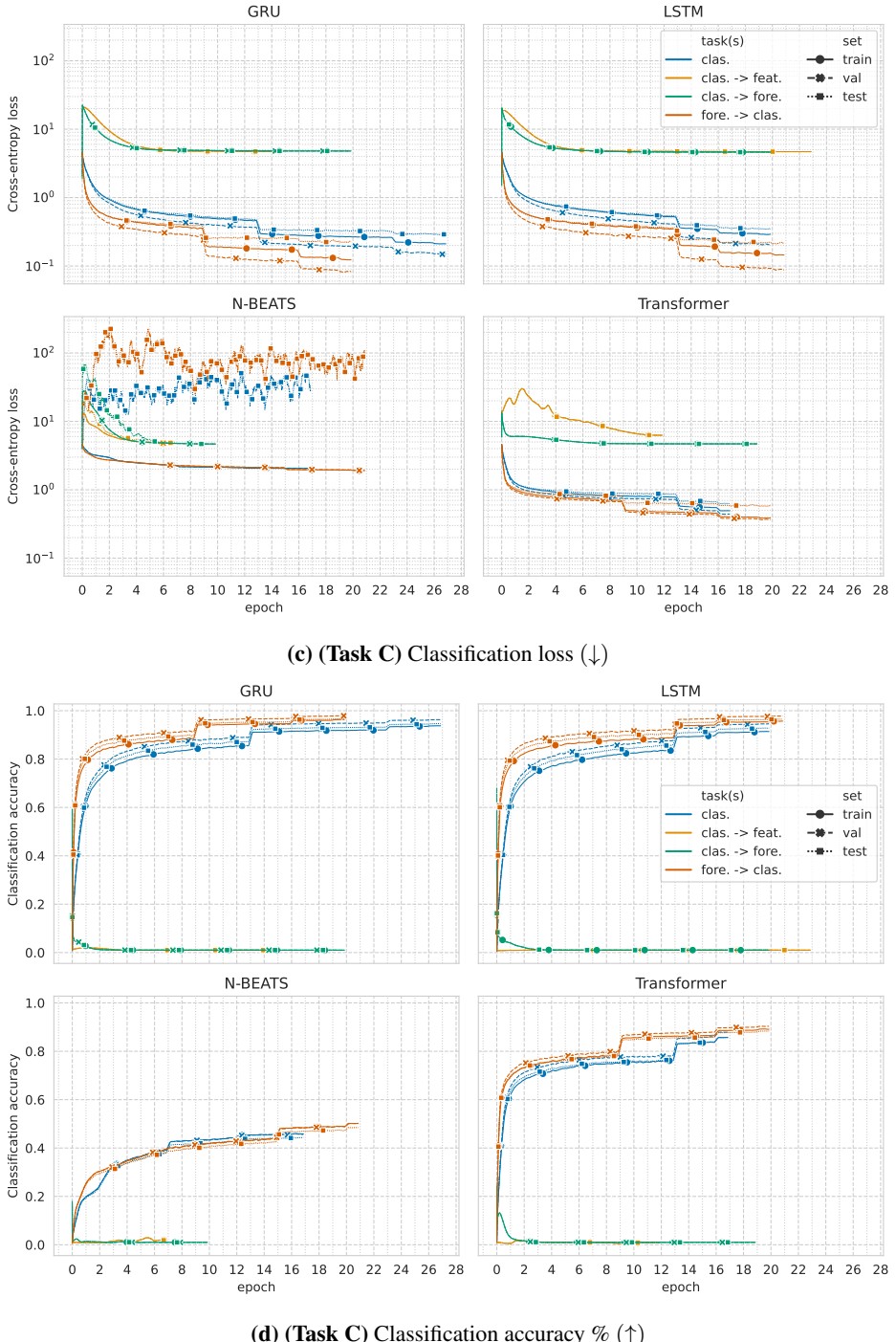

**(c) (Task C)** Classification loss ($\downarrow$)

**(d) (Task C)** Classification accuracy % ($\uparrow$)

**Figure A4:** Prober experiment: training curves (cont.)

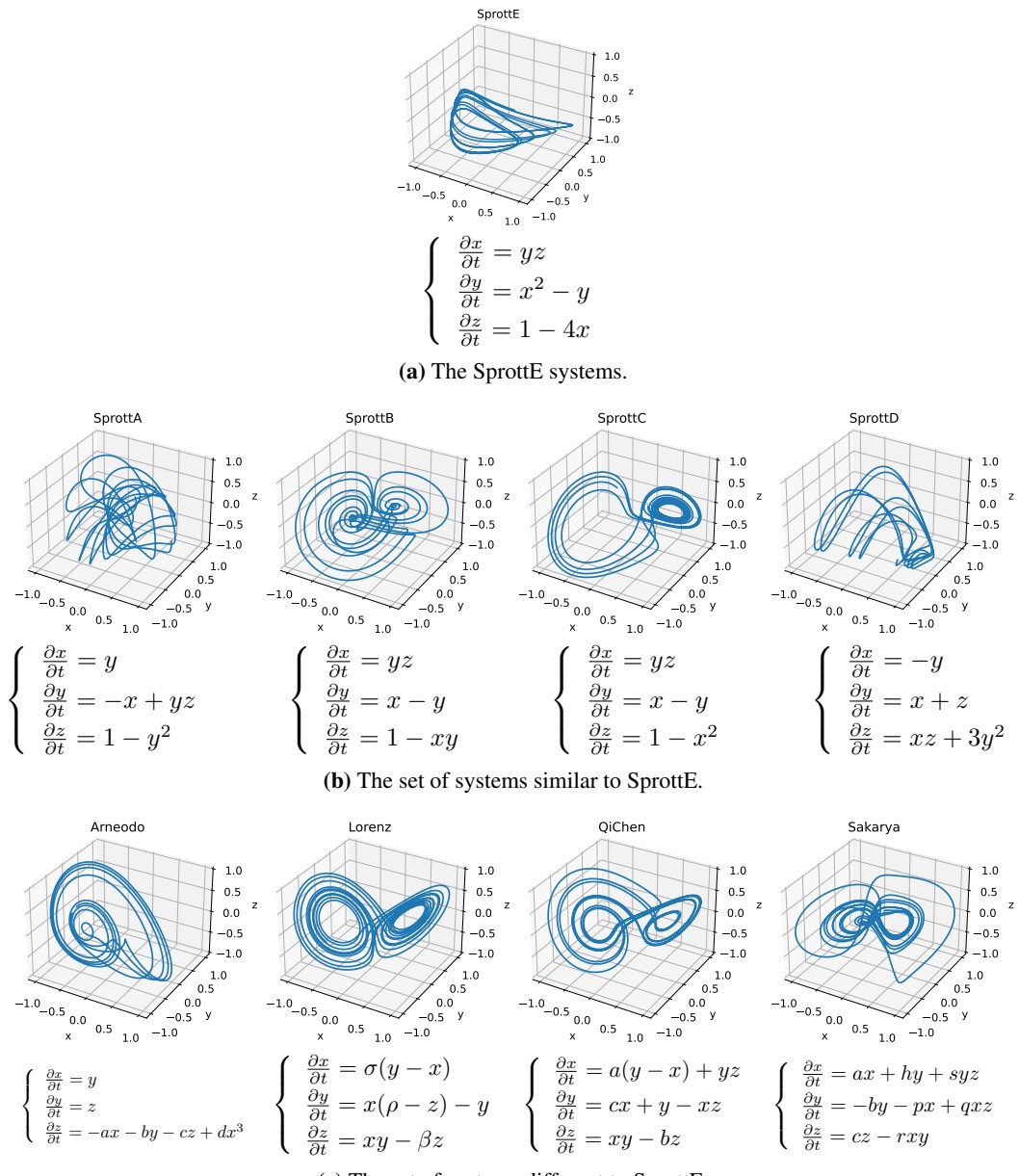

**Figure A5:** Few-shot learning experiment: The set of 9 systems used in the experiment. The default trajectory of 500 points per period and 10 periods is shown for each. Each trajectory component is re-scaled to be in the range [-1, 1]. The differential equations of SprottE and the similar systems are simpler, with at most a sum of two elementary products, while those of the different systems involve more complicated terms.

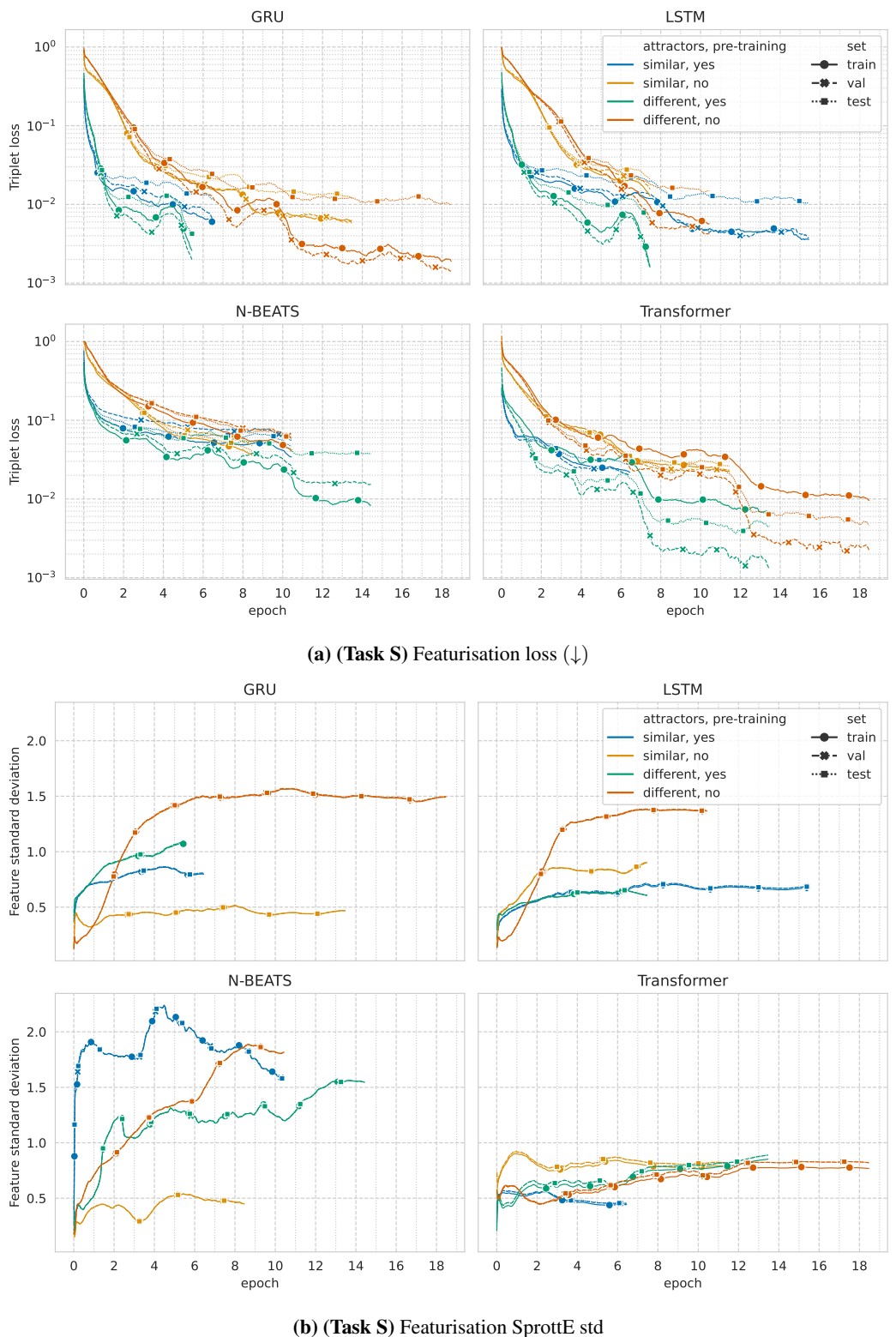

**(a) (Task S)** Featurisation loss ($\downarrow$)

**(b) (Task S)** Featurisation SprottE std

**Figure A6:** Few-shot learning experiment: Supervised Featurization: training curves (only runs with SprottE are included). A running average of length 100 (roughly an epoch) is used for readability.
All runs with pre-training converge faster except those of N-BEATS and classification sensitivity.

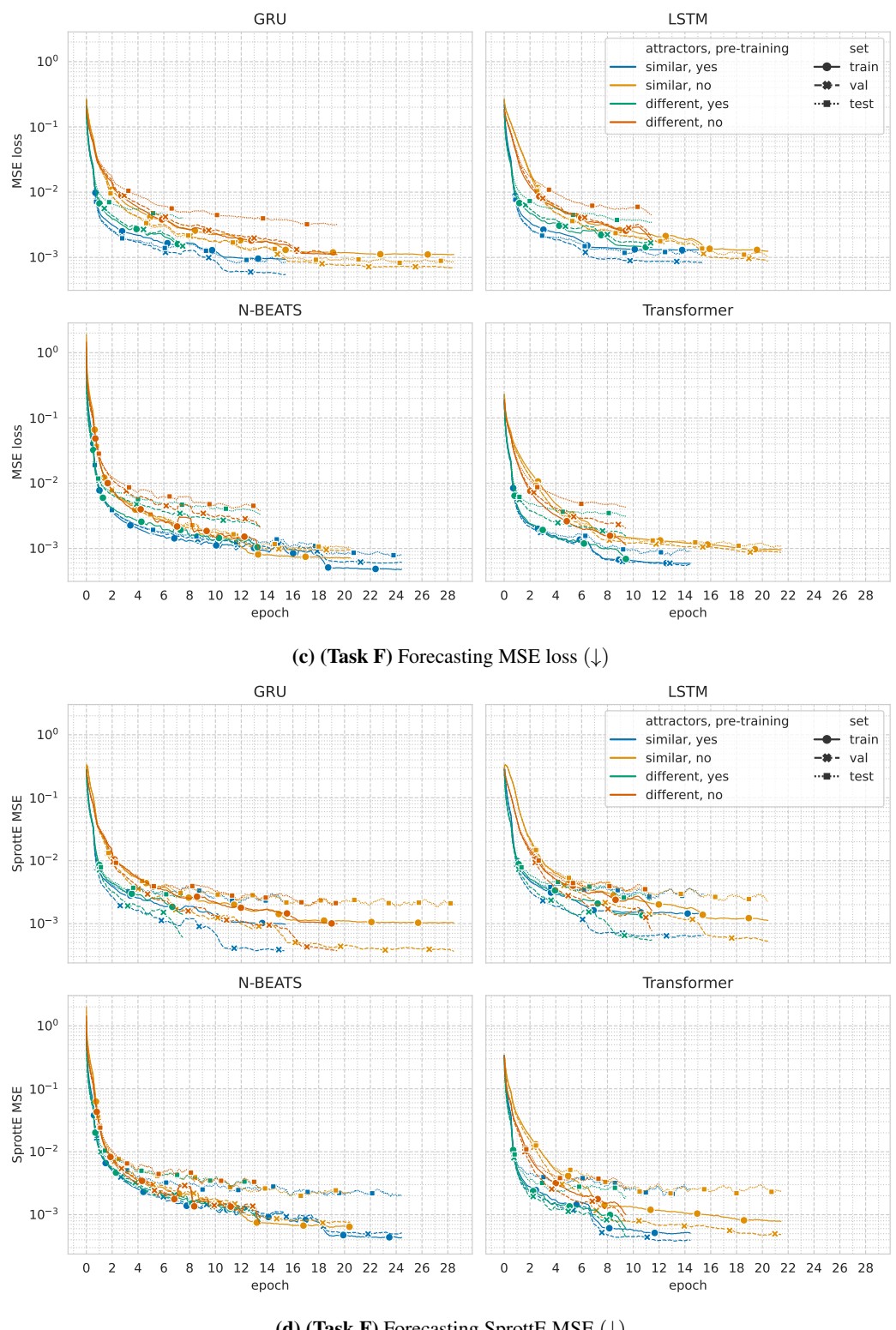

**(c) (Task F)** Forecasting MSE loss ($\downarrow$)

**(d) (Task F)** Forecasting SprottE MSE ($\downarrow$)

**Figure A6:** Few-shot learning experiment: Forecasting: training curves (only runs with SprottE are included). A running average of length 100 (roughly an epoch) is used for readability.
All runs with pre-training converge faster except those of N-BEATS and classification sensitivity.

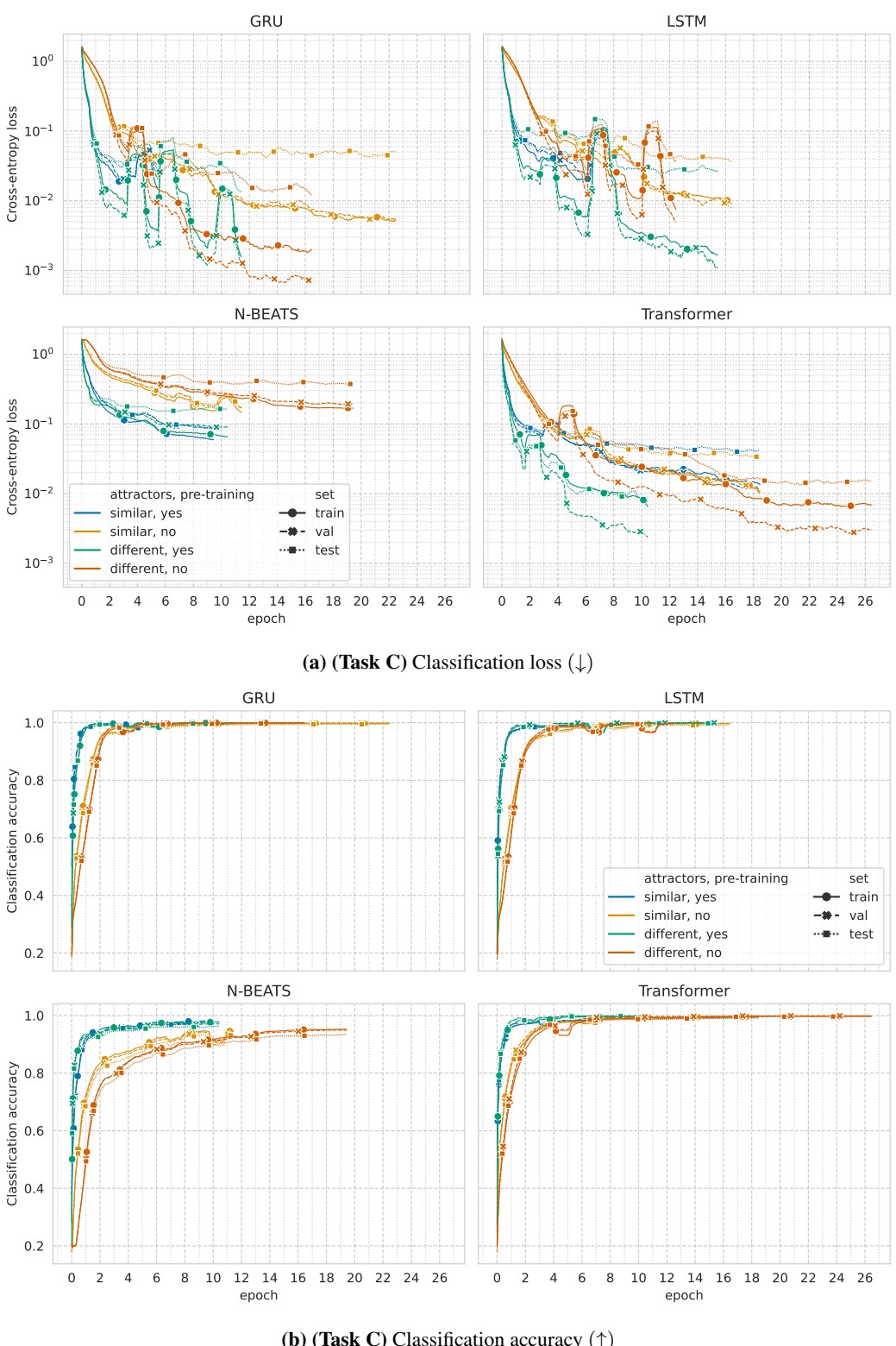

**(a) (Task C)** Classification loss (↓)

**(b) (Task C)** Classification accuracy (↑)

**Figure A7:** Few-shot learning experiment: Classification loss, accuracy: training curves (only runs with SprottE are included). A running average of length 100 (roughly an epoch) is used for readability.
All runs with pre-training converge faster except those of N-BEATS and classification sensitivity.

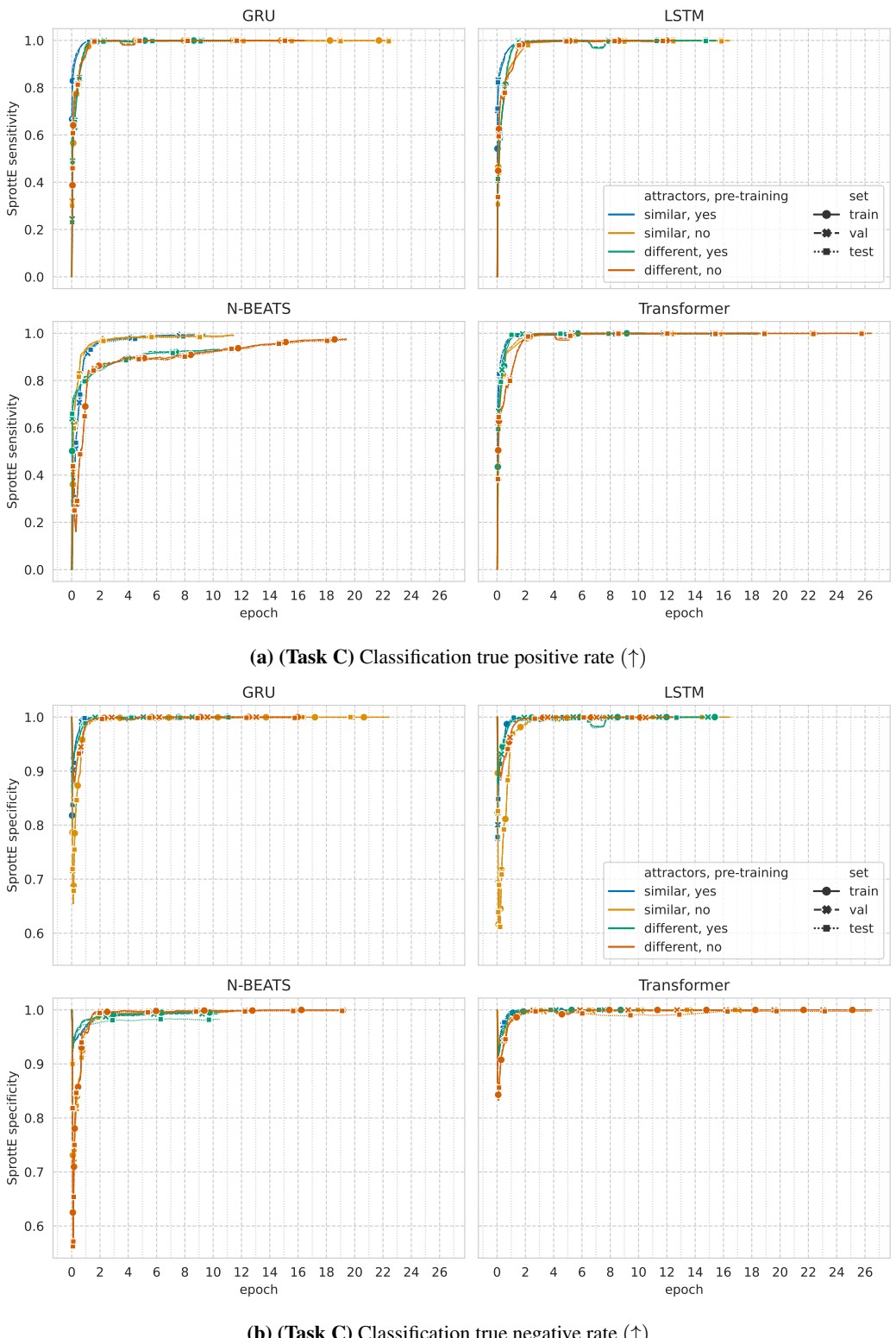

**(a) (Task C)** Classification true positive rate ($\uparrow$)

**(b) (Task C)** Classification true negative rate ($\uparrow$)

**Figure A8:** Few-shot learning experiment: Classification TPR, TNR: training curves (only runs with SprottE are included). A running average of length 100 (roughly an epoch) is used for readability. All runs with pre-training converge faster except those of N-BEATS and classification sensitivity.

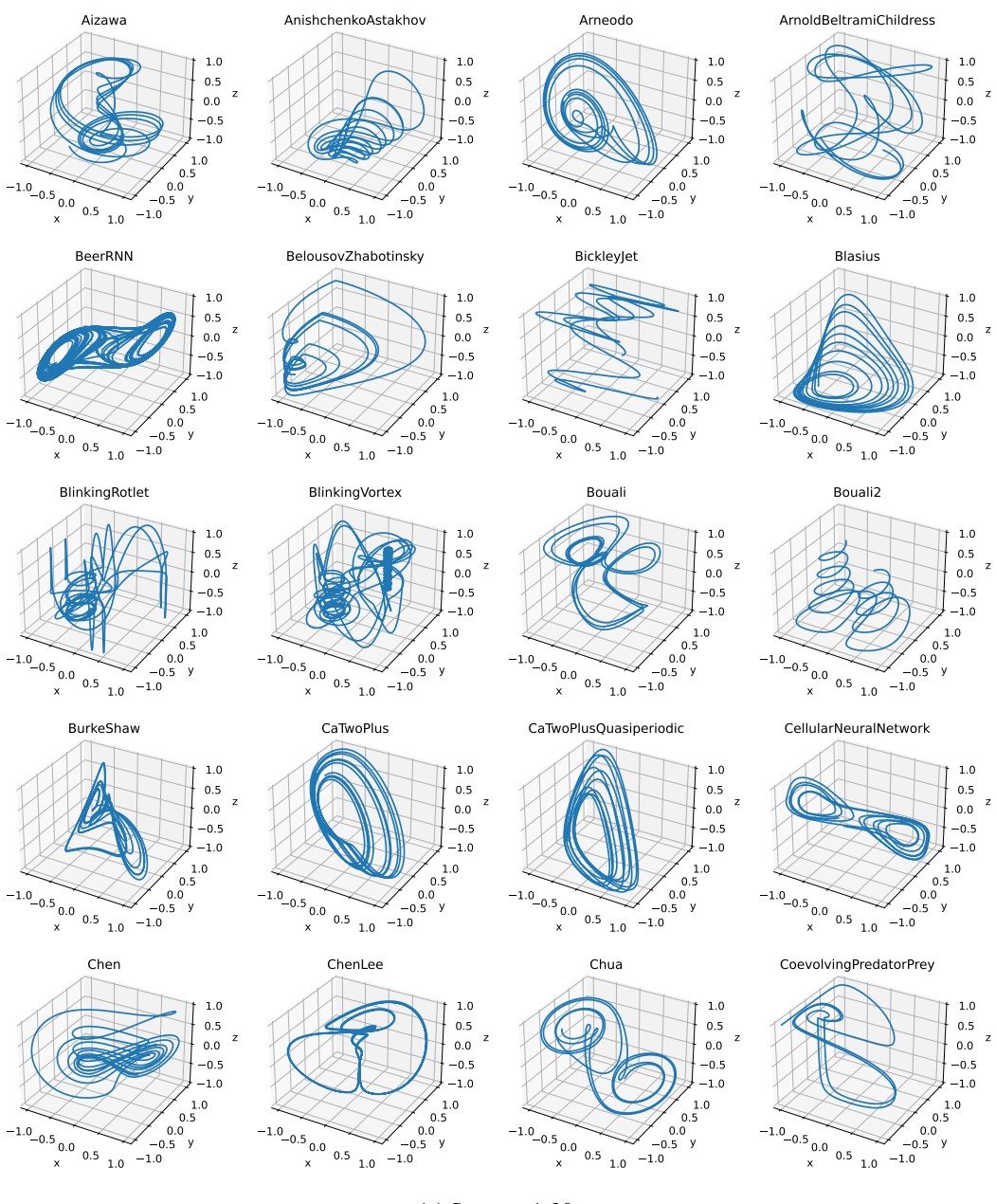

(a) Systems 1-20.

**Figure A9:** The 100 dynamical systems of dimension 3 in our synthetic dataset (only Torus is non-chaotic). A single trajectory is shown for each, with the default initial condition, 500 points per period, and 10 periods. Each trajectory component is re-scaled to be in the range $[-1, 1]$.

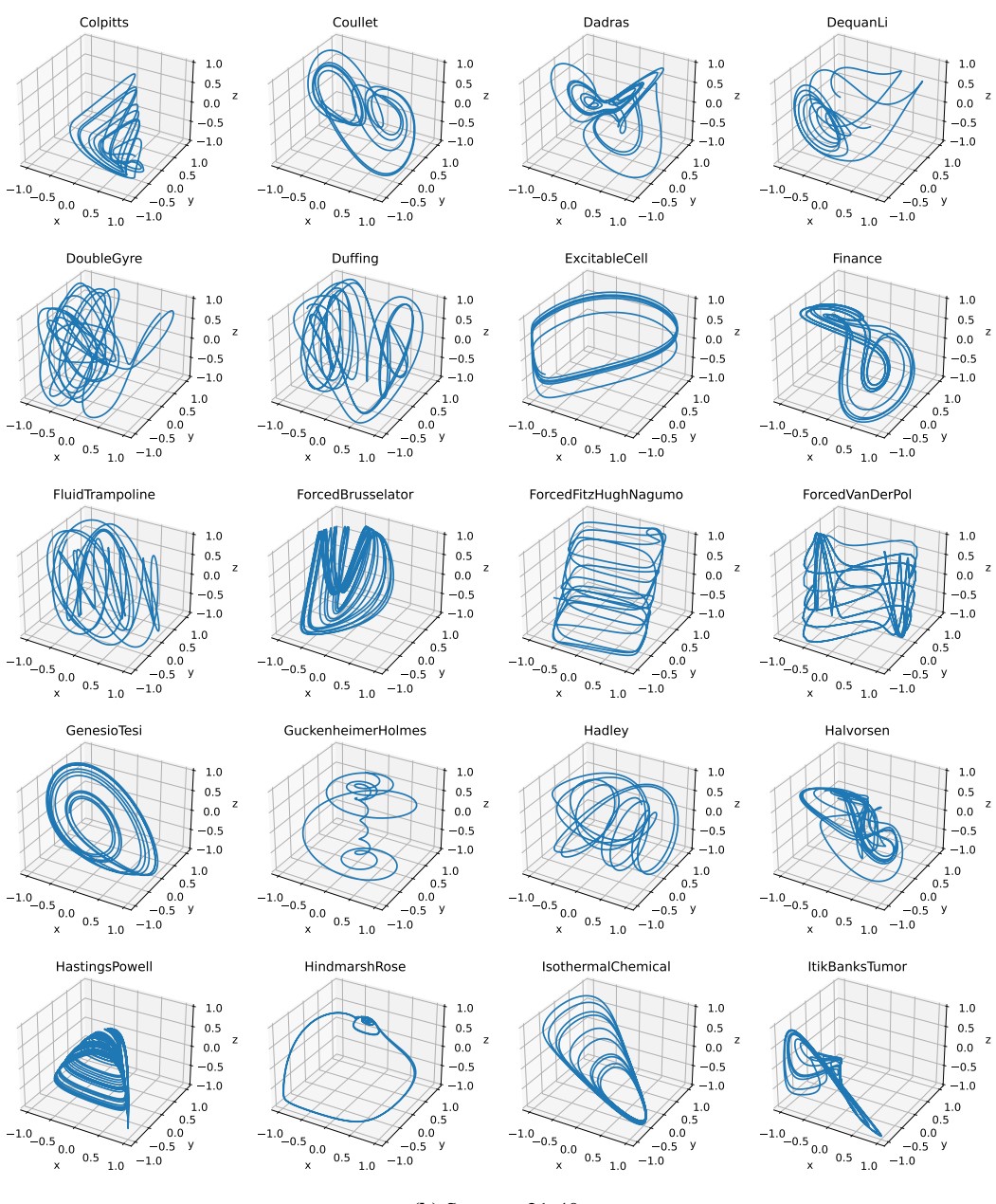

**(b)** Systems 21-40.

**Figure A9:** The 100 chaotic dynamical systems of dimension 3. (cont.)

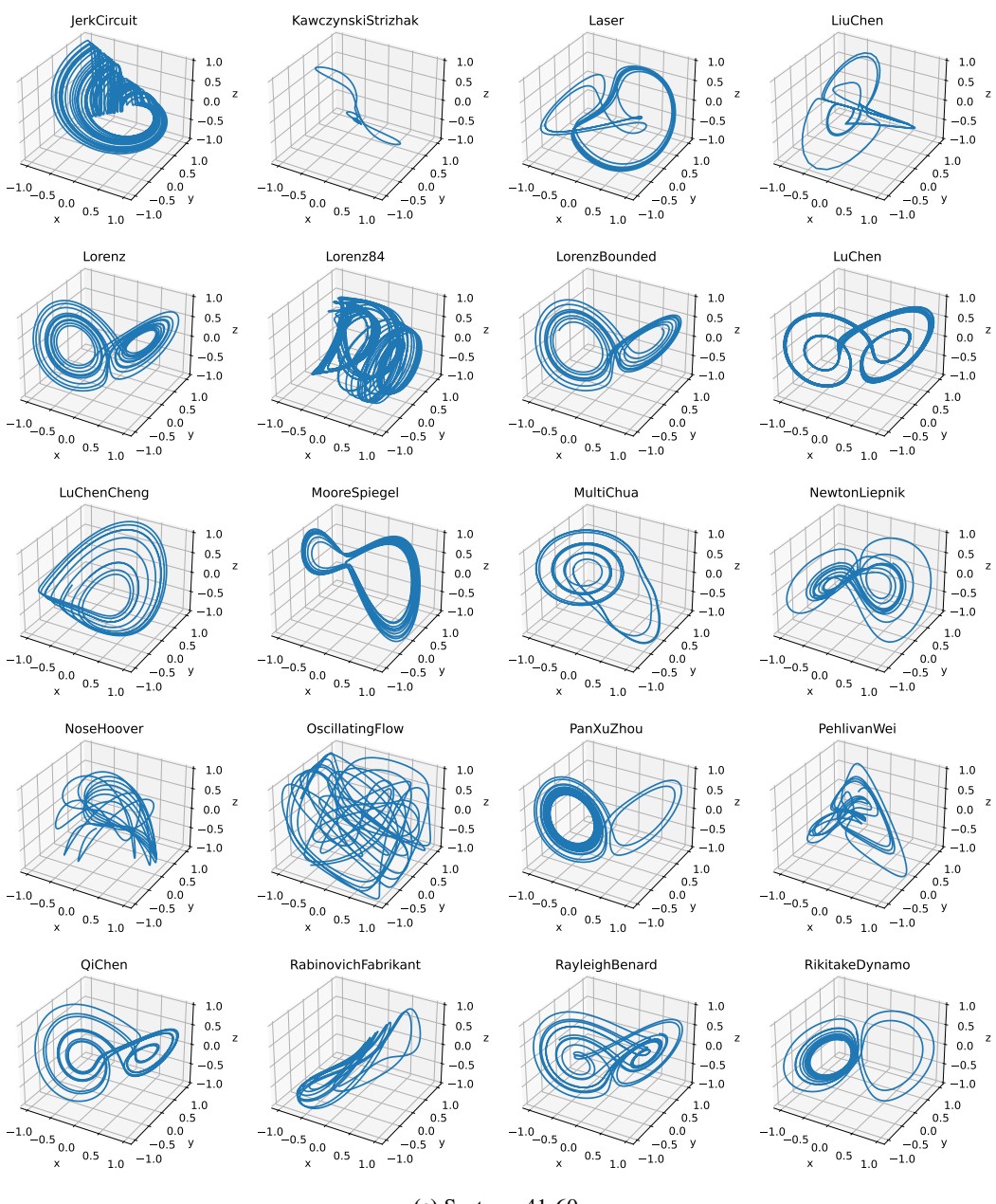

(c) Systems 41-60.

**Figure A9:** The 100 chaotic dynamical systems of dimension 3. (cont.)

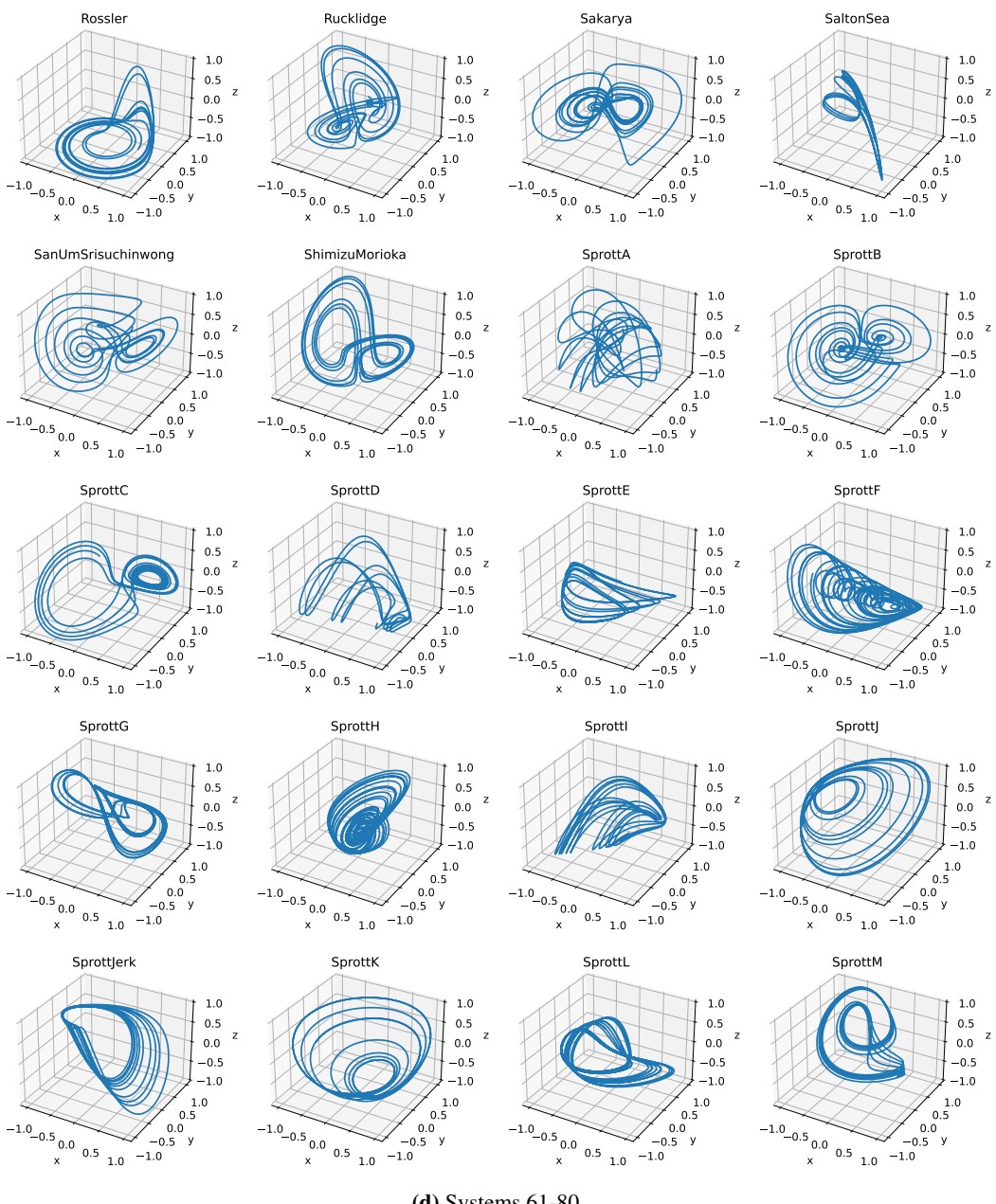

**(d)** Systems 61-80.

**Figure A9:** The 100 chaotic dynamical systems of dimension 3. (cont.)

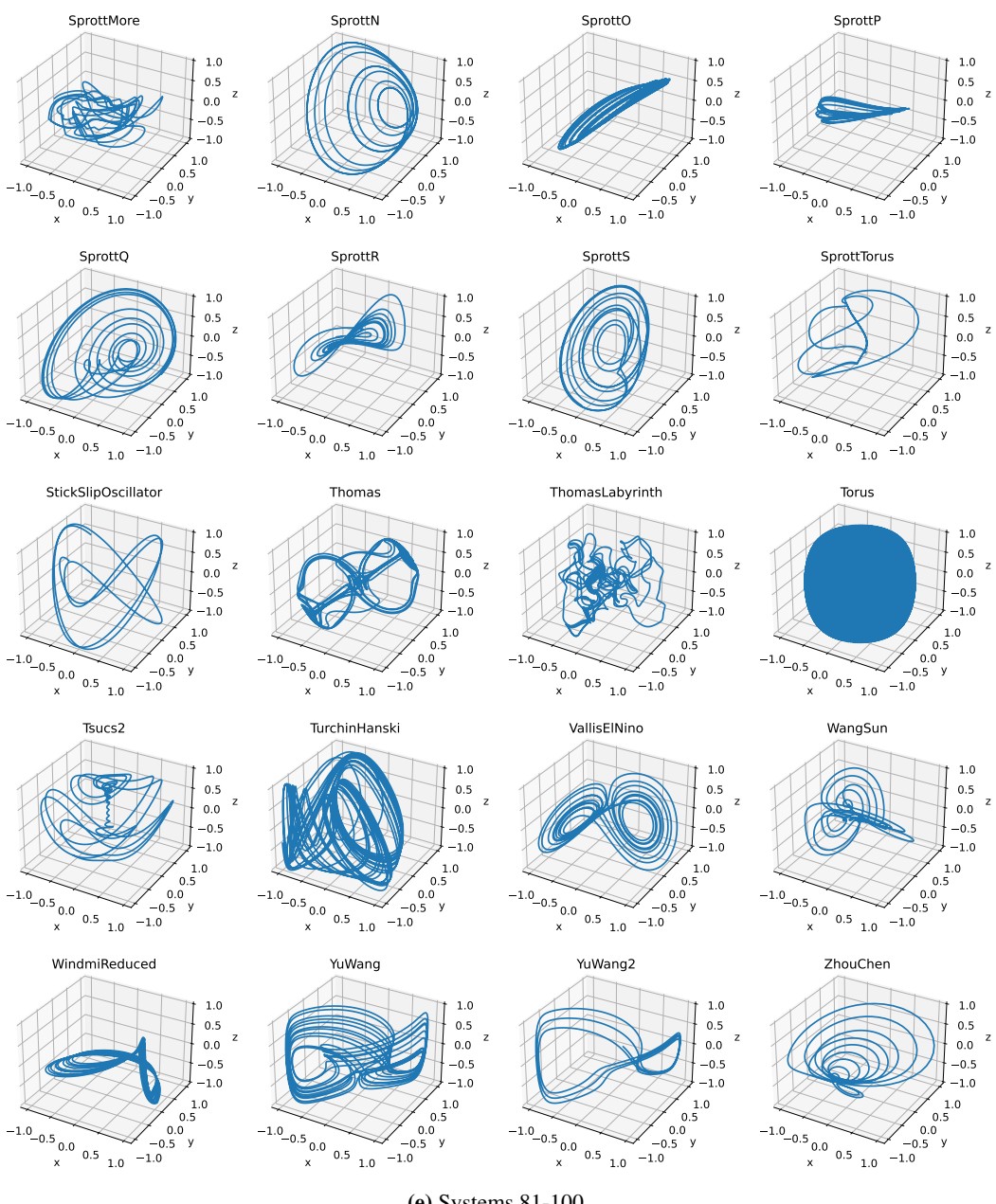

**(e)** Systems 81-100.

**Figure A9:** The 100 chaotic dynamical systems of dimension 3. (cont.)

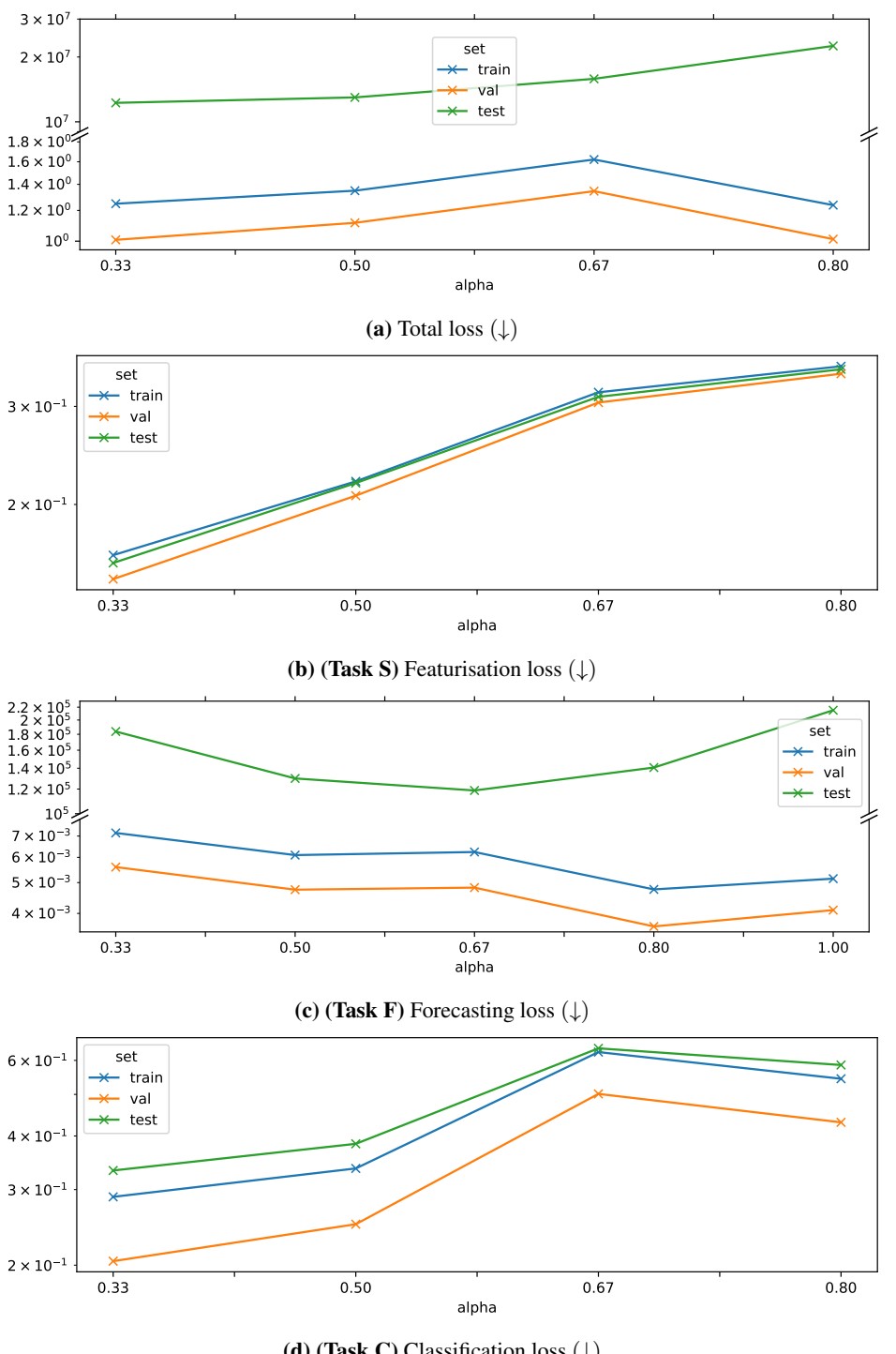

**(a)** Total loss ($\downarrow$)

**(b) (Task S)** Featurisation loss ($\downarrow$)

**(c) (Task F)** Forecasting loss ($\downarrow$)

**(d) (Task C)** Classification loss ($\downarrow$)

**Figure A10:** Multi-task loss experiment: final GRU performance for different $\alpha$ (MSE weight in $L_{\text{total}}$).

The forecasting train/val loss decreases with $\alpha$, showing that GRU does not benefit from the enforced system representations from the other tasks. The featurisation and classification losses generally increase with $\alpha$, so they do not benefit from the forecasting representations either.

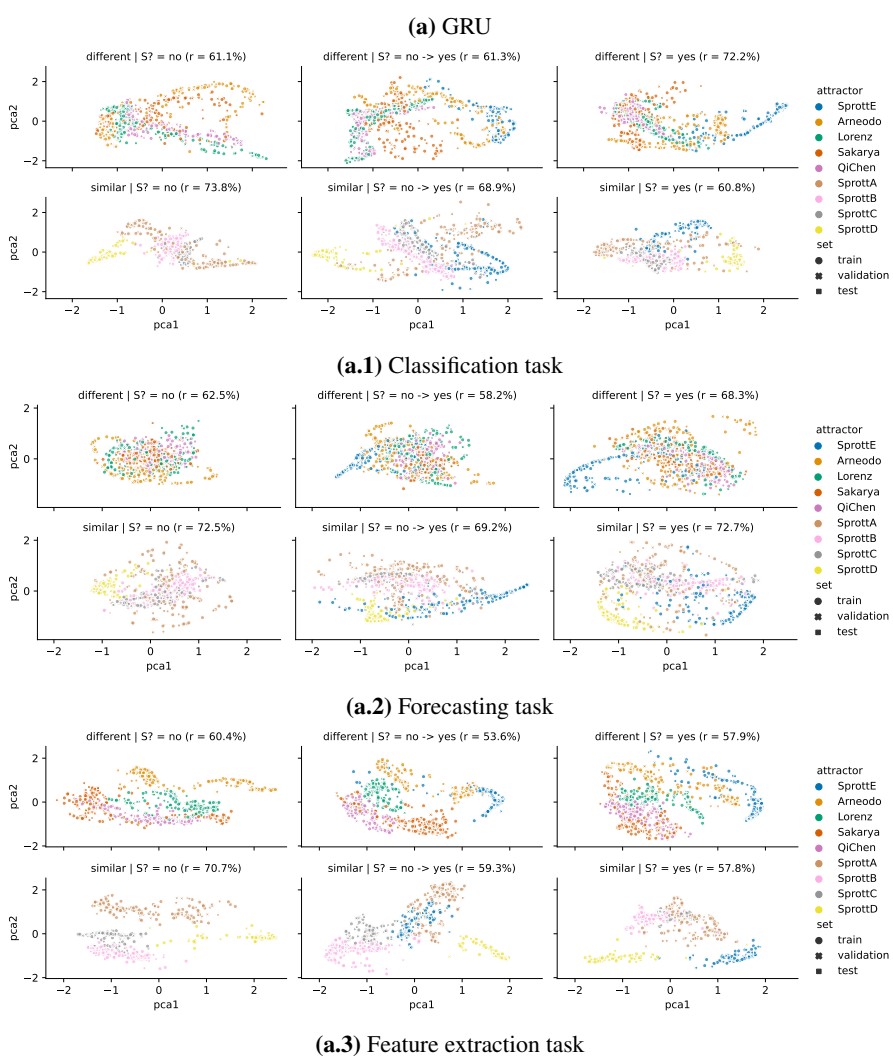

**Figure A11:** Few-shot learning experiment: PCA projection of the features learned by the models as a function of the other attractors used in the experiment (similar or different), and whether the SprottE attractor is included (shown as 'S?'). A different PCA is performed for each model run, using the 32-dimensional features obtained for the train/validation/test datasets, and we show the percentage of variance captured by the two main PCA axes as 'r'.

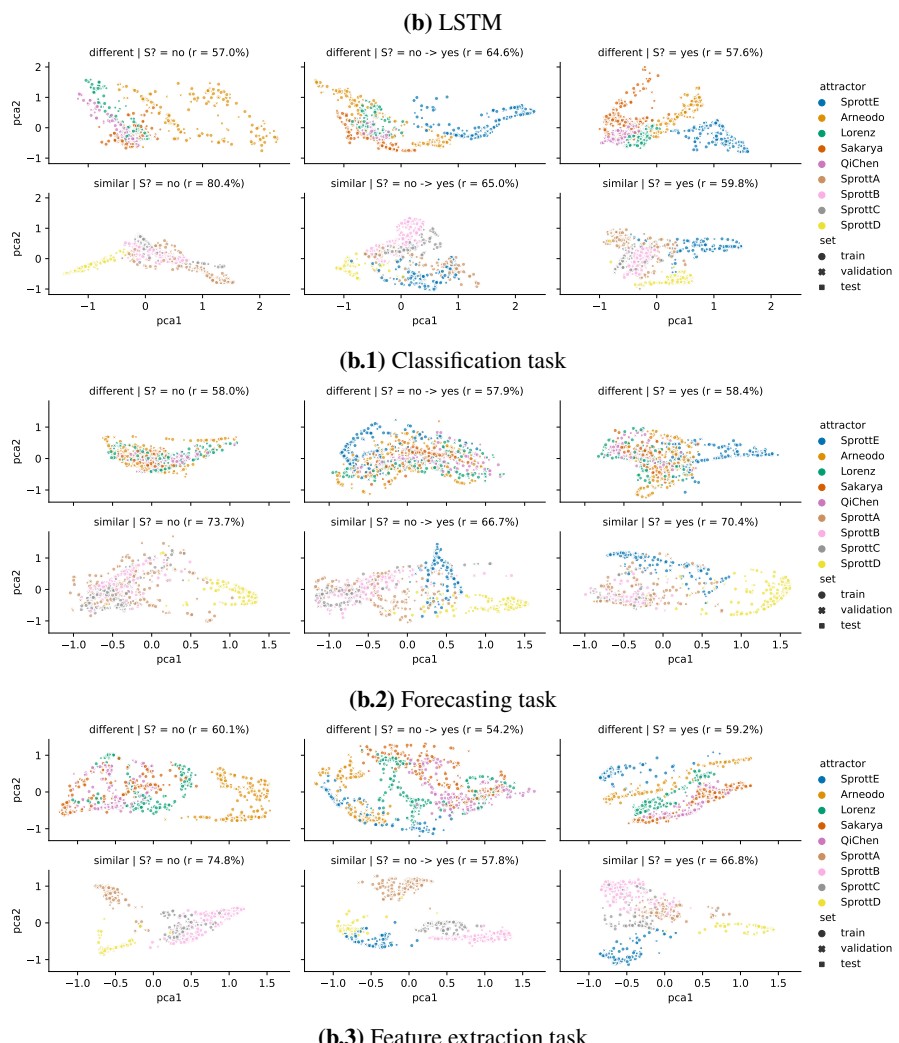

**Figure A11:** Few-shot learning experiment: PCA projection of learned features (cont.)

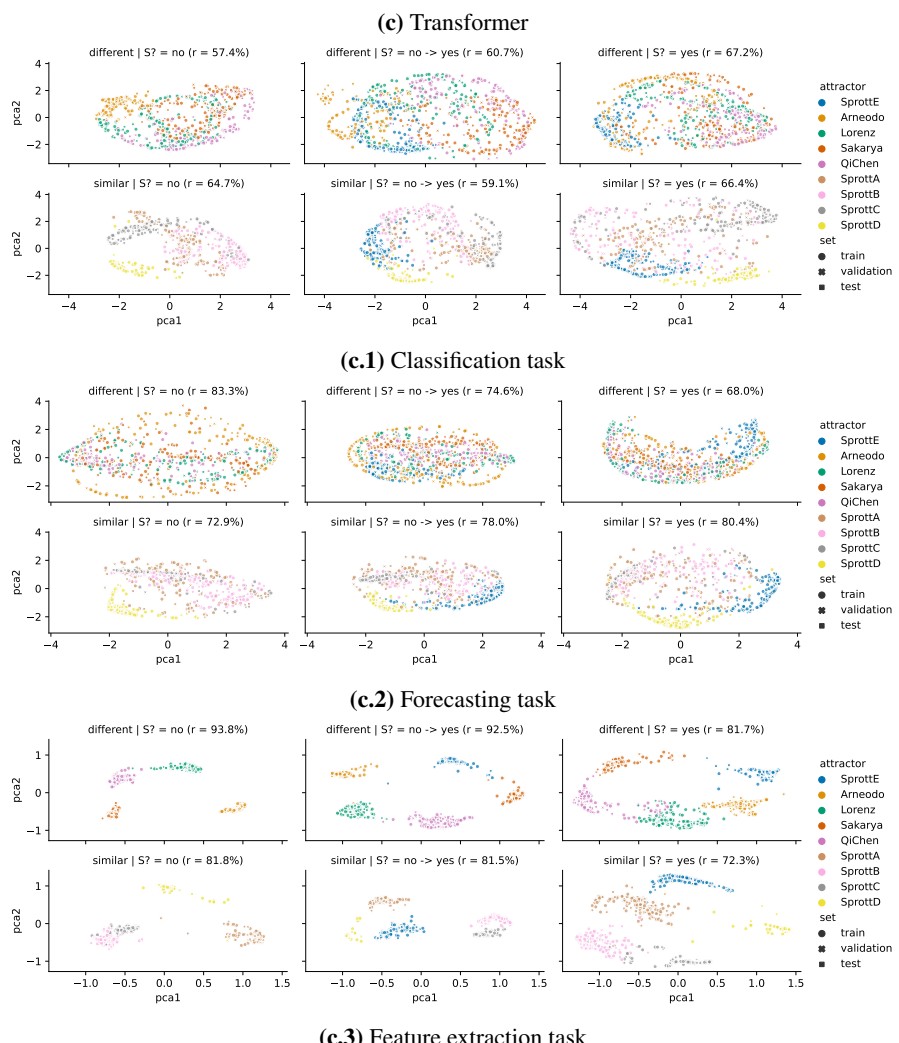

**Figure A11:** Few-shot learning experiment: PCA projection of learned features (cont.)

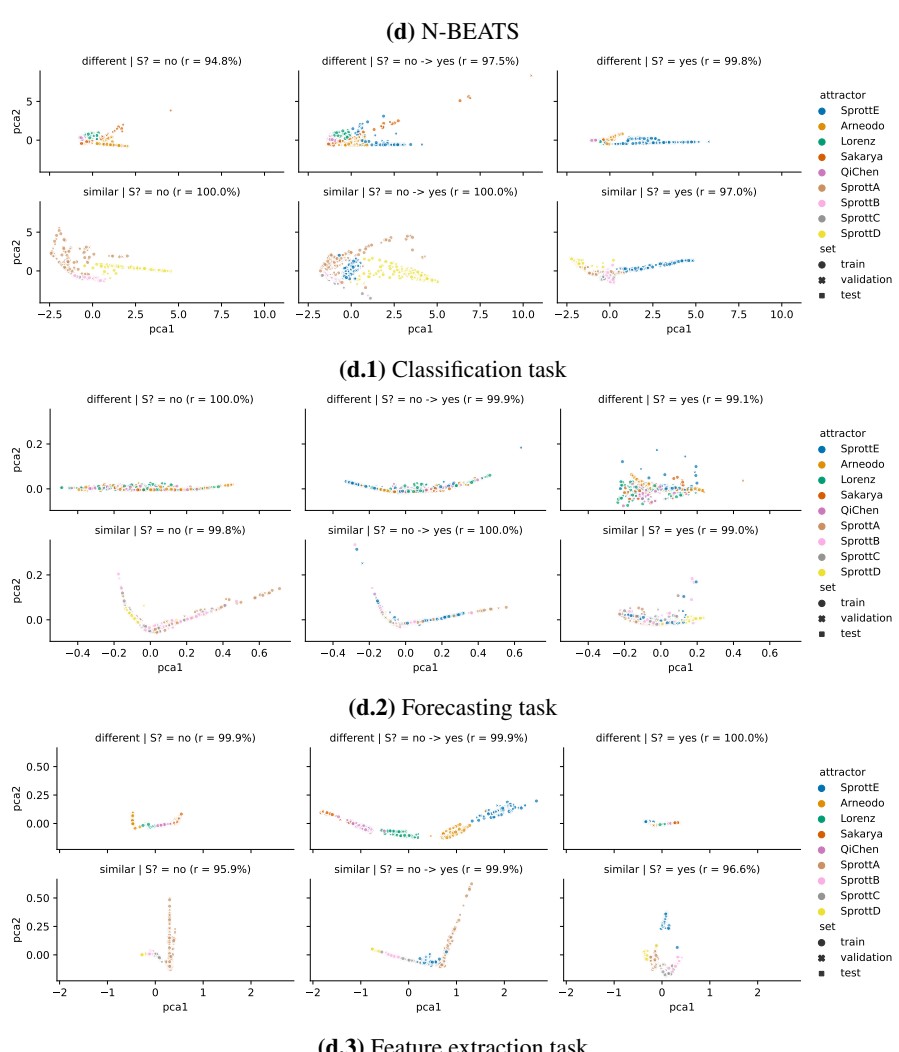

**Figure A11:** Few-shot learning experiment: PCA projection of learned features (cont.)

