# OpenReview forum: "Factors Influencing Generalization in Chaotic Dynamical Systems"
_ICLR.cc/2023/Conference — Submitted to ICLR 2023_

### Official Review · Reviewer_k5uo · 2022-10-17

**Confidence:** 5
**Correctness:** 2
**Technical Novelty And Significance:** 2
**Empirical Novelty And Significance:** 2
**Recommendation:** 3

**Clarity, Quality, Novelty And Reproducibility:**

Clarity:
* The paper would greatly benefit from restructuring the experimental analysis around a particular hypothesis, and then lining up all pieces of evidence that support it.

Novelty:
* To my knowledge, a complete analysis of transfer learning for chaotic systems has not been carried out in the literature. I would suggest the authors focus on the analysis and not the particular framework, which is not technically novel or motivated against any other time series framework.

Reproducibility:
* The results are not reproducible, code and details are not provided.

**Strength And Weaknesses:**

Strengths:
* Understanding transfer in deep time series models  (from a theoretical or even purely empirical perspective) remains an open problem in the literature, and this paper promises a step forward in that direction.
* I appreciate the design of certain experiments, in particular finding the probing metric idea valuable. Some of the findings are potentially interesting, but as will be made clear below this paper suffers from lack of focus and depth in the analysis which in my opinion makes me lose confidence in the findings.

Concerns (high-level, clarity):
* Please argue with facts when presenting your contribution (e.g. "The framework saves a large amount of code
repetition and complex indexing/references, e.g. in multi-task experiments." is a weird way to start your method section, especially when these statements are not contextualized e.g., w.r. to what? how much is saved?)
* I don't understand why the "framework" is presented as the core technical contribution. What makes this different than any other time series paper doing forecasting or classification or self-supervision?. As this is primarily an empirical investigation, I would have personally preferred more details on the experiments themselves rather than attempts to justify the novelty of performing classification, forecasting and self-supervised training on time series in a single repository. I understand some implementation choices have been made to carry out the experiments, leading to some abstractions (the diagram in A1), but I am not convinced these are any better or easier to use than just working in PyTorch Lightning directly, and the authors do not spend time justifying their choices. The end result is a paper that is uncomfortably in-between a systems paper and an empirical analysis paper, with no justification for the abstraction and no convincing takeaways from the empirical analysis.

Concerns (experimental):
* Elaborating on the above point. There is a surprising lack of motivation for anything attempted in this paper e.g., (1) why have these four model classes been chosen? If the focus was covering as many mainstream deep time series models, the list is missing temporal convolutions, deep state space models, neural differential equations and more. If the focus was on identifying a particular property of a model class, the paper misses the mark as there is no analysis, nor any ablation to attempt to validate their observations. For example: you observe transfer from forecasting to classification in Transformers? Show that this holds across datasets, varying the architecture, attempt to motivate why you think that is. In my opinion, few readers will be convinced by the evidence you presented in support of your findings, especially due to the large number of moving pieces and the large scope of the paper, which leaves little space for depth. The authors might want to see [1] for a completely empirical work that puts forth a hypothesis, verifies it and then provides further compelling evidence for it. (2) What makes any of this analysis tailored to chaotic systems, beyond the title and choice of datasets? There is no attempt to ablate the empirical findings against a dataset of non-chaotic or turbulent systems.
* The low performance of N-BEATS seems to go against a lot of time series literature, and I cannot check the implementation since the code was not provided. From the loss divergence seen in Figure A2 and others it appears the hyperparameters might have been poorly chosen.
* If classification accuracy reaches ~100% for all models, it is a sign that perhaps a more challenging task or variation of the task is required. There is no signal here to distinguish models.


[1] ResNet strikes back: An improved training procedure in timm

**Summary Of The Paper:**

This work presents an empirical investigation of generalization and transfer in learning to predict dynamical systems. A set of experiments involving classification, featurization and forecasting for 4 main model types is implemented in a framework the authors call ValiDyna.

**Summary Of The Review:**

The authors carry out an investigation of performance of deep time series models for forecasting, classification and self-supervised training on small-dimensional chaotic systems. Some preliminary empirical findings are provided (e.g., freezing after pertaining does not work, some degree of transfer from forecasting to classification, or from forecasting to featurisation). The experiments are bundled with a high-level framework built on PyTorch Lightning. The paper would benefit from a narrower scope and more depth in the analysis.

---

### Official Review · Reviewer_KoFr · 2022-10-21

**Confidence:** 4
**Correctness:** 2
**Technical Novelty And Significance:** 2
**Empirical Novelty And Significance:** 2
**Recommendation:** 3

**Clarity, Quality, Novelty And Reproducibility:**

This paper tackles an interesting question, generalization in dynamical systems. However, it does so with so little background in dynamical systems theory, the current state of the field or relevant model architectures for learning dynamical systems, that the results are, in my mind, hardly useful.

In more detail:

1) There is a burgeoning and rich literature on learning dynamical systems in ML and physics meanwhile, which was almost completely ignored here (with less than a handful of relevant citations). For a start see https://arxiv.org/abs/2207.00521, https://arxiv.org/abs/2110.07238, https://arxiv.org/abs/2010.08895, https://arxiv.org/abs/1802.07486, https://arxiv.org/abs/2006.13431,  https://arxiv.org/pdf/2201.05136, https://arxiv.org/abs/1710.07313, https://openreview.net/pdf?id=vwj6aUeocyf, https://arxiv.org/abs/2207.02542,  https://arxiv.org/pdf/2202.07022, https://arxiv.org/abs/1712.09707, https://openreview.net/pdf?id=aUX5Plaq7Oy.

2) Perhaps as a result of this, no architectures or training protocols designed for learning dynamical systems were tested. All models tested taken in their vanilla form would not be expected to perform well, as addressed in the previous literature on this topic. This may explain some of the poor generalization or learning results observed in this paper.

3) Also, the previous literature discussed in quite some detail suitable metrics for judging the test performance on dynamical systems. For instance, the cited Gilpin paper collects many of them, including things like measures of fractal dimensionality, topological features, Lyapunov exponents, or geometrical state space correlation (see also https://arxiv.org/abs/2207.02542, https://arxiv.org/abs/2207.00521, for instance). None of these were assessed, and therefore the question remains how indicative the measures chosen by the authors really are for the training and generalization success of any model.

4) The authors claim they test out-of-distribution (OOD) generalization, but the only experiment that partially does so is the one in sect. 5.4. In all other experiments, in my understanding, simply new trajectories (initial conditions) were drawn from the same system with same parameters within the same domain or invariant set. So they are expected to have the same statistics (and no proof otherwise, theoretical or empirical, was provided anyway). As a starter on this topic of how to test generalization for dynamical systems, perhaps consult https://arxiv.org/abs/2207.00521 and the literature cited in there.

5) Formal precision and detail is lacking in many sections. For instance, in sect. 5.4 benchmark models were classified as ‘similar’ or ‘different’ based on obscure criteria. For instance, *all* the ODE systems included in this sect. are given by three 2nd (or 3rd) order polynomials, so not sure on which grounds the authors classify some of these as more simple than or more similar to others. More crucially, the mathematical form of these equations is much less relevant here than the geometry and topology of the dynamics, e.g. number of holes. The same equations may produce profoundly different dynamics in different parameter regimes.
Some other examples (not an exhaustive list):
- unclear what a,p,n are in 4.1 and why this def. is useful
- unclear how initialization for forecasting was precisely done
- some of the systems used (A.9) are apparently non-chaotic but exhibit limit cycles
- p.2, bottom: Motivation of generation process unclear; if the used ODE solver exhibits numerical instabilities, then switch to an implicit and/or adaptive step-size solver (solvers used were also not mentioned).
- SEMs were not provided in many of the tables

6) Since this study is only concerned with 3-dimensional noise-free systems, one may also question the relevance to real world problems as those mentioned in the beginning. This is another aspect where the present study falls behind the current state in the literature where real-world and high-dimensional, noisy, partially observed, systems were considered as well.

7) In general, I found the present study hardly adds anything new to the existing literature (much of which, as noted above, was not covered in the related works). Much better generalization procedures, based on more useful metrics for dynamical systems, have been designed in the literature listed above and references therein. No new models or relevant training protocols were developed, no new theoretical insights or framework, no novel metrics or test sets.

**Strength And Weaknesses:**

Strength:
- addresses an interesting \& highly relevant question

Weaknesses:
- does actually not really test OOD generalization, as claimed
- doesn't consider relevant benchmarks from the lit.
- misses important tests for generalization

**Summary Of The Paper:**

The authors construct a framework for assessing generalization performance of models for learning dynamical systems. For this they collect a variety of simulated benchmark data sets and introduce different tasks (forecasting, generalization, classification), and different evaluation measures and procedures. They exemplify their framework with LSTM, GRU, Transformer and N-Beats.

**Summary Of The Review:**

The question of generalization is certainly very important, but this paper in my mind does not address it. It lacks the theoretical and empirical background necessary to do so.

---

### Official Review · Reviewer_hEVM · 2022-10-22

**Confidence:** 3
**Clarity, Quality, Novelty And Reproducibility:** The main idea of the paper sounds novel.
**Correctness:** 2
**Technical Novelty And Significance:** 2
**Empirical Novelty And Significance:** 2
**Recommendation:** 5

**Strength And Weaknesses:**


**Weaknesses**:

1) Some parts of the paper are unclear. For example, regarding the sentence "*We sample data from each dynamical system by picking different initial conditions. This leads to trajectories that are sufficiently different from each other, but [...]*" (on page 2, Section 3.1), there are the following questions:
- What does "sufficiently different" mean in this sentence? Both *sufficiently* and *different* need to be clarified.

- How does picking different initial conditions lead to *qualitatively* different trajectories for a dynamical system? For instance, suppose that one chooses different initial conditions from the basin of attraction of an attractor. Then, all the generated trajectories can be qualitatively the same (topologically equivalent).

2) If possible, it would be better to increase the size of some plots (e.g. the plots on pages 31-34 which are not clear enough).

**Summary Of The Paper:**

The purpose of this paper is to present an experimental analysis of factors that influence generalization for data exhibiting chaotic dynamics. For this purpose, the authors built a configurable and extensible model evaluation framework (Validyna). A number of experiments were constructed and run using Validyna.

**Summary Of The Review:**

overall, I think the paper's main idea is substantial, and interesting; but needs technical improvements.

---

### Decision · Program_Chairs · 2023-01-20

**Decision:**

Reject

**Justification For Why Not Higher Score:**

N/A

**Justification For Why Not Lower Score:**

N/A

**Metareview: Summary, Strengths And Weaknesses:**

The authors did not respond to many criticisms raised by the reviewers.
I recommend rejection.

**Summary Of Ac-Reviewer Meeting:**

N/A